# A Scale-Invariant Sorting Criterion to Find a Causal Order in Additive Noise Models

**Alexander G. Reisach**[*]
CNRS, MAP5
Université Paris Cité
F-75006 Paris, France

**Myriam Tami**
CentraleSupélec
Université Paris-Saclay
F-91190 Gif-sur-Yvette, France

**Christof Seiler**
Department of Advanced
Computing Sciences, Maastricht
University, The Netherlands

**Antoine Chambaz**
CNRS, MAP5
Université Paris Cité
F-75006 Paris, France

**Sebastian Weichwald**
Department of Mathematical Sciences
and Pioneer Centre for AI,
University of Copenhagen, Denmark

## Abstract

Additive Noise Models (ANMs) are a common model class for causal discovery from observational data. Due to a lack of real-world data for which an underlying ANM is known, ANMs with randomly sampled parameters are commonly used to simulate data for the evaluation of causal discovery algorithms. While some parameters may be fixed by explicit assumptions, fully specifying an ANM requires choosing all parameters. Reisach et al. (2021) show that, for many ANM parameter choices, sorting the variables by increasing variance yields an ordering close to a causal order and introduce 'var-sortability' to quantify this alignment. Since increasing variances may be unrealistic and cannot be exploited when data scales are arbitrary, ANM data are often rescaled to unit variance in causal discovery benchmarking.

We show that synthetic ANM data are characterized by another pattern that is scale-invariant and thus persists even after standardization: the explainable fraction of a variable's variance, as captured by the coefficient of determination $R^2$, tends to increase along the causal order. The result is high '$R^2$-sortability', meaning that sorting the variables by increasing $R^2$ yields an ordering close to a causal order. We propose a computationally efficient baseline algorithm termed '$R^2$-SortnRegress' that exploits high $R^2$-sortability and that can match and exceed the performance of established causal discovery algorithms. We show analytically that sufficiently high edge weights lead to a relative decrease of the noise contributions along causal chains, resulting in increasingly deterministic relationships and high $R^2$. We characterize $R^2$-sortability on synthetic data with different simulation parameters and find high values in common settings. Our findings reveal high $R^2$-sortability as an assumption about the data generating process relevant to causal discovery and implicit in many ANM sampling schemes. It should be made explicit, as its prevalence in real-world data is an open question. For causal discovery benchmarking, we provide implementations of $R^2$-sortability, the $R^2$-SortnRegress algorithm, and ANM simulation procedures in our library CausalDisco.

---

[*]Correspondence to alexander.reisach@math.cnrs.fr

37th Conference on Neural Information Processing Systems (NeurIPS 2023).

# 1 Introduction

**Causal Discovery in Additive Noise Models.** Understanding the causal relationships between variables is a common goal across the sciences (Imbens and Rubin 2015) and may facilitate reliable prediction, the identification of goal-directed action, and counterfactual reasoning (Spirtes, Glymour, and Scheines 1993; Pearl 2009; Peters, Janzing, et al. 2017). Causal reasoning using Structural Causal Models (SCMs) consists of discovering a graph encoding the causal structure between variables, learning the corresponding functions of the SCM, and using them to define and estimate causal quantities of interest. Discovering causal structure requires interventional data or assumptions on the functions and distributions to restrict the class of possible SCMs that describe a given data-generating process (see Glymour et al. 2019). Since collecting interventional data may be costly, infeasible, or unethical, we are interested in suitable assumptions to learn causal structure from observational data. Additive Noise Models (ANMs) encode a popular functional assumption that yields identifiability of the causal structure under various additional assumptions on the noise distributions, given the causal Markov (Kiiveri et al. 1984) and faithfulness (Spirtes, Glymour, and Scheines 1993) assumptions. In ANMs, the causal graph can be identified by testing the independence of regression residuals (see Hoyer et al. 2008; Mooij et al. 2009; Peters, Mooij, et al. 2011). Another approach is to use assumptions on the relative magnitudes of noise variances for identifiability (Bühlmann et al. 2014; Loh and Bühlmann 2014; Ghoshal and Honorio 2018; Chen et al. 2019; Park 2020). A range of causal discovery algorithms use such assumptions to find candidate causal structures (for an overview see Heinze-Deml et al. 2018; Kitson et al. 2023). Well-known structure learning algorithms include the constraint-based *PC* algorithm (Spirtes and Glymour 1991) and the score-based fast greedy equivalence search (*FGES*) algorithm (Meek 1997; Chickering 2002). Using a characterization of directed acyclic graphs as level set of a differentiable function over adjacency matrices paved the way for causal discovery employing continuous optimization (Zheng et al. 2018), which has inspired numerous variants (Vowels et al. 2022).

**Implicit Assumptions in Synthetic ANM Data Generation.** The use of synthetic data from simulated ANMs is common for the evaluation of causal discovery algorithms because there are few real-world datasets for which an underlying ANM is known with high confidence. For such benchmark results to be indicative of how causal discovery algorithms (and estimation algorithms, see Curth et al. 2021) may perform in practice, simulated data need to plausibly mimic observations of real-world data generating processes. Reisach et al. (2021) demonstrate that data in ANM benchmarks exhibit high var-sortability, a measure quantifying the alignment between the causal order and an ordering of the variables by their variances (see Section 3 for the definition). High var-sortability implies a tendency for variances to increase along the causal order. As a consequence, sorting variables by increasing variance to obtain a causal order achieves state-of-the-art results on those benchmarks, on par with or better than those of continuous structure learning algorithms, the *PC* algorithm, or *FGES*. Var-sortability also explains why the magnitude of regression coefficients may contain more information about causal links than the corresponding p-values (Weichwald et al. 2020). However, an increase in variances along the causal order may be unrealistic because it would lead to downstream variables taking values outside plausible ranges. On top of that, var-sortability cannot be exploited on real-world data with arbitrary data scales because variances depend on the data scale and measurement unit. For these reasons, following the findings of Reisach et al. (2021), Kaiser and Sipos (2022), and Seng et al. (2022), synthetic ANM data are often standardized to control the increase of variances (see, for example, Rios et al. 2021; Rolland et al. 2022; Mogensen et al. 2022; Lorch et al. 2022; Xu et al. 2022; Li et al. 2023; Montagna et al. 2023), which deteriorates the performance of algorithms that rely on information in the variances. As first demonstrated for var-sortability (Reisach et al. 2021), the parameter choices necessary for sampling different ANMs constitute implicit assumptions about the resulting data-generating processes. These assumptions may result in unintended patterns in simulated data, which can strongly affect causal discovery results. Post-hoc standardization addresses the problem only superficially and breaks other scale-dependent assumptions on the data generating process, such as assumptions on the magnitude of noise variances. Moreover, standardization does not alter any scale-independent patterns that may inadvertently arise in ANM simulations. This motivates our search for a scale-invariant criterion that quantifies such scale-independent patterns in synthetic ANM data. Assessing and controlling their impact on causal discovery helps make ANM simulations more realistic and causal discovery benchmark performances more informative.

**Contribution.** We show that in data from ANMs with parameters randomly sampled following common practice, not only variance, but also the fraction of variance explainable by the other variables, tends to increase along the causal order. We introduce $R^2$-sortability to quantify the alignment between a causal order and the order of increasing coefficients of determination $R^2$. In $R^2$-SortnRegress, we provide a baseline algorithm exploiting $R^2$-sortability that performs well compared to established methods on common simulation-based causal discovery benchmarks. In $R^2$, we propose a sorting criterion for causal discovery that is applicable even when the data scale is arbitrary, as is often the case in practice. Unlike var-sortability, one cannot simply rescale the obtained variables to ensure a desired $R^2$-sortability in the simulated ANMs. Since real-world $R^2$-sortabilities are yet unknown, characterizing $R^2$-sortability and uncovering its role as a driver of causal discovery performance enables one to distinguish between data with different levels of $R^2$-sortability when benchmarking or applying causal discovery algorithms to real-world data. To understand when high $R^2$-sortability may be expected, we analyze the emergence of $R^2$-sortability in ANMs for different parameter sampling schemes. We identify a criterion on the edge weight distribution that determines the convergence of $R^2$ to $1$ along causal chains. An empirical analysis of random graphs confirms that the weight distribution strongly affects $R^2$-sortability, as does the average in-degree. The ANM literature tends to focus on assumptions on the additive noise distributions (see for example Shimizu et al. 2006; Peters and Bühlmann 2014; Park 2020), leading to arguably arbitrary choices for the other ANM parameters not previously considered central to the model class (for example, drawing edge weights iid from some distribution). Our analysis contributes to closing the gap between theoretical results on the identifiability of ANMs and the need to make decisions on all ANM parameters in simulations. Narrowing this gap is necessary to ensure that structure learning results on synthetic data are meaningful and relevant in practice.

## 2 Additive Noise Models

We follow standard practice in defining the model class of linear ANMs. Additionally, we detail the sampling process to generate synthetic data.

Let $X = [X_1, ..., X_d]^\top \in \mathbb{R}^d$ be a vector of $d$ random variables. The causal structure between the components of $X$ can be represented by a graph over nodes $\{X_1, ..., X_d\}$ and a set of directed edges between nodes. We encode the set of directed edges by a binary adjacency matrix $B \in \{0, 1\}^{d \times d}$ where $B_{s,t} = 1$ if $X_s \to X_t$, that is, if $X_s$ is a direct cause of $X_t$, and $B_{s,t} = 0$ otherwise. Throughout, we assume every graph to be a directed acyclic graph (DAG). Based on a causal DAG, we define an ANM. Let $\sigma = [\sigma_1, \ldots, \sigma_d]^\top$ be a vector of positive iid random variables. Let $\mathcal{P}_N(\phi)$ be a distribution with parameter $\phi$ controlling the standard deviation. Given a draw $\boldsymbol{\sigma}$ of noise standard deviations, let $N = [N_1, ..., N_d]^\top$ be the vector of independent noise variables with $N_t \sim \mathcal{P}_N(\boldsymbol{\sigma}_t)$. For each $t$, let $f_t$ be a measurable function such that $X_t = f_t(\mathbf{Pa}(X_t)) + N_t$, where $\mathbf{Pa}(X_t)$ is the set of parents of $X_t$ in the causal graph. We assume that $f_1, ..., f_d$ and $\mathcal{P}_N(\phi)$ are chosen such that all $X_1, ..., X_d$ have finite second moments and zero mean. Since we can always subtract empirical means, the assumption of zero means does not come with any loss of generality; in our implementations we subtract empirical means or use regression models with an intercept.

In this work, we consider linear ANMs and assume that $f_1, \ldots, f_d$ are linear functions. For linear ANMs, we define a weight matrix $W \in \mathbb{R}^{d \times d}$ where $W_{s,t}$ is the weight of the causal link from $X_s$ to $X_t$ and $W_{s,t} = 0$ if $X_s$ is not a parent of $X_t$. The structural causal model can be written as

$$X = W^\top X + N. \tag{1}$$

### 2.1 Synthetic Data Generation

ANMs with randomly sampled parameters are commonly used for generating synthetic data and benchmarking causal discovery algorithms (see for example Scutari 2010; Ramsey et al. 2018; Kalainathan et al. 2020). We examine a pattern in data generated from such ANMs ($R^2$-sortability, see Section 3.1) and design an algorithm to exploit it ($R^2$-SortnRegress, see Section 3.2). Here, we describe the steps to sample ANMs for synthetic data generation using the following parameters:

| Parameter | Description |
|---|---|
| $d$ | Number of nodes |
| $P_{\mathcal{G}}$ | Graph structure distribution |
| $\gamma$ | Average node in-degree |
| $\mathcal{P}_N(\phi)$ | Parametrized noise distribution |
| $P_\sigma$ | Distribution of noise standard deviations |
| $P_W$ | Distribution of edge weights |

**1. Generate the Graph Structure.** We generate the causal graph by sampling a DAG adjacency matrix $B \in \{0,1\}^{d \times d}$ for a given number of nodes $d$. In any DAG, the variables can be permuted such that its adjacency matrix is upper triangular. We use this fact to obtain a random DAG adjacency matrix from undirected random graph models by first deleting all but the upper triangle of the adjacency matrix and then shuffling the variable order (Zheng et al. 2018). In our simulations, we use the Erdős–Rényi (ER) (Erdős, Rényi, et al. 1960) and scale-free (SF) (Barabási and Albert 1999) random graph models for $P_{\mathcal{G}}$ with an average node in-degree $\gamma$. We denote the distribution of Erdős–Rényi random graphs with $d$ nodes and $\gamma d$ edges as $\mathcal{G}_{\mathrm{ER}}(d, \gamma d)$, and the distribution of scale-free graphs with the same parameters as $\mathcal{G}_{\mathrm{SF}}(d, \gamma d)$.

**2. Define Noise Distributions.** In linear ANMs, each variable $X_t$ is a linear function of its parents plus an additive noise variable $N_t$. We choose a distributional family for $\mathcal{P}_N(\phi)$ (for example, zero-mean Gaussian or Uniform) and independently sample standard deviations $\boldsymbol{\sigma}_1, ..., \boldsymbol{\sigma}_d$ from $P_\sigma$. We then set $N_t$ to have distribution $\mathcal{P}_N(\boldsymbol{\sigma}_t)$ with standard deviation $\boldsymbol{\sigma}_t$.

**3. Draw Weight Parameters.** We sample $\beta_{s,t}$ for $s, t = 1, ..., d$ independently from $P_W$ and define the weight matrix $W = B \odot [\beta_{s,t}]_{s,t=1,...,d}$, where $\odot$ denotes the element-wise product.

**4. Sample From the ANM.** To sample observations from the ANM with given graph (step 1), edge weights (step 2), and noise distributions (step 3), we use that $\mathrm{Id} - W^\top$ is invertible if $W$ is the adjacency matrix of a DAG to re-arrange Equation (1) and obtain the data-generating equation:

$$X = \left(\mathrm{Id} - W^\top\right)^{-1} N.$$

We denote a dataset of $n$ observations of $X$ as $\mathbf{X} = \left[\mathbf{X}^{(1)}, ..., \mathbf{X}^{(n)}\right]^\top \in \mathbb{R}^{n \times d}$. The $i$-th observation of variable $t$ in $\mathbf{X}$ is $\mathbf{X}_t^{(i)}$, and $\mathbf{X}^{(i)} \in \mathbb{R}^d$ is the $i$-th observation vector.

## 3 Defining and Exploiting $R^2$-Sortability

We describe a pattern in the fractions of explainable variance along the causal order and propose to measure it as '$R^2$-sortability'. $R^2$-sortability is closely related to var-sortability (Reisach et al. 2021), which measures a pattern in the variances of variables in a causal graph that can be exploited for causal discovery. Similarly to var-sortability, $R^2$-sortability can also be exploited to learn the causal structure of ANMs by use of a simple algorithm. In contrast to var-sortability, $R^2$-sortability is invariant to rescaling of the variables and the presented algorithm, $R^2$-*SortnRegress*, recovers the causal structure equally well from raw, standardized, or rescaled data.

### 3.1 From Var-Sortability to $R^2$-Sortability

Var-sortability measures the agreement between an ordering by variance and a causal ordering. In many common ANM simulation schemes, the variances of variables tend to increase along the causal order, leading to high var-sortability. This accumulation of noise along the causal order motivates our investigation of a related pattern. The variance of a variable $X_t$ is given as

$$\mathrm{Var}(X_t) = \mathrm{Var}(W_{:,t}^\top X) + \mathrm{Var}(N_t) = \mathrm{Var}(W_{:,t}^\top X) + \sigma_t^2.$$

If the variance $\mathrm{Var}(X_t)$ increases for different $X_t$ along the causal order, but each noise variance $\sigma_t^2$ is sampled iid, then the fraction of cause-explained variance

$$\frac{\mathrm{Var}(W_{:,t}^\top X)}{\mathrm{Var}(W_{:,t}^\top X) + \sigma_t^2}$$

is also likely to increase for different $X_t$ along the causal order. Unlike the variance, we cannot calculate (nor obtain an unbiased estimate of) a variable's fraction of cause-explained variance without knowing its parents in the graph. Instead, we propose to use an upper bound as a proxy for the fraction of cause-explained variance by computing the fraction of explainable variance of a variable $X_t$ given all remaining variables, known as the coefficient of determination

$$R^2 = 1 - \frac{\text{Var}\left(X_t - \mathbb{E}\left[X_t \mid X_{\{1,\dots,d\}\setminus\{t\}}\right]\right)}{\text{Var}\left(X_t\right)}$$

(Glantz et al. 2001). In practice, we need to choose a regression model and regress the variable onto all other variables to estimate this quantity. Here, we choose linear models $M_{t,S}^{\theta} \colon \mathbb{R}^{|S|} \to \mathbb{R}, X_S \mapsto \langle \theta, X_S \rangle$ for the regression of $X_t$ onto $X_S$ with $S \subseteq \{1,\dots,d\} \setminus \{t\}$ and $\theta \in \mathbb{R}^{|S|}$. We denote the least-squares fit by $M_{t,S}^{\theta^*}$ and estimate the $R^2$ coefficient of determination of $X_t$ given $X_S$ via

$$R^2(M_{t,S}^{\theta^*}, X) = 1 - \frac{\text{Var}\left(X_t - M_{t,S}^{\theta^*}(X_S)\right)}{\text{Var}\left(X_t\right)}, \quad \text{for } S = \{1,\dots,d\} \setminus \{t\}.$$

Per definition, $R^2$ is scale-invariant: it does not change when changing the scale of components of $X$.

To find a common definition for sortability by different ordering criteria, we introduce a family of $\tau$-sortability criteria $\mathbf{v}_\tau$ that, for different functions $\tau$, assign a scalar in $[0,1]$ to the variables $X$ and graph $G$ (with adjacency matrix $B_{\mathcal{G}}$) of an ANM as follows:

$$\mathbf{v}_\tau(X,\mathcal{G}) = \frac{\sum_{i=1}^{d} \sum_{(s\to t)\in B_{\mathcal{G}}^i} \text{incr}(\tau(X,s),\tau(X,t))}{\sum_{i=1}^{d} \sum_{(s\to t)\in B_{\mathcal{G}}^i} 1} \quad \text{where incr}(a,b) = \begin{cases} 1 & a < b \\ 1/2 & a = b \\ 0 & a > b \end{cases} \quad (2)$$

and $B_{\mathcal{G}}^i$ is the $i$-th matrix power of the adjacency matrix $B_{\mathcal{G}}$ of graph $\mathcal{G}$, and $(s \to t) \in B_{\mathcal{G}}^i$ if and only if at least one directed path from $X_s$ to $X_t$ of length $i$ exists in $\mathcal{G}$. In effect, $\mathbf{v}_\tau(X,\mathcal{G})$ is the fraction of directed paths of unique length between any two nodes in $\mathcal{G}$ satisfying the sortability condition in the numerator. Note that other definitions of sortability that count paths differently are possible, see Appendix D for a comparison to two alternative definitions. We obtain the original var-sortability for $\tau(X,t) = \text{Var}\left(X_t\right)$ and denote it as $\mathbf{v}_{\text{Var}}$. We obtain $R^2$-sortability for $\tau(X,t) = R^2(M_{t,\{1,\dots,d\}\setminus\{t\}}^{\theta^*}, X)$ and denote it as $\mathbf{v}_{R^2}$.

If $\mathbf{v}_{R^2}$ is 1, then the causal order is identified by the variable order of increasing $R^2$ (decreasing if $\mathbf{v}_{R^2} = 0$). A value of $\mathbf{v}_{R^2} = 0.5$ means that ordering the variables by $R^2$ amounts to a random guess of the causal ordering. $R^2$-sortability is related but not equal to var-sortability, and an ANM that may be identifiable under one criterion may not be so under the other. We present a quantitative comparison across simulation settings in Appendix C, showing that when $R^2$-sortability is high, it tends to be close to var-sortability (as well as to a sortability by cause-explained variances). The correspondence may be arbitrarily distorted by re-scaling of the variables since var-sortability is scale sensitive, whereas $R^2$-sortability is scale-invariant. In the following subsection we show that high $R^2$-sortability can be exploited similarly to high var-sortability.

## 3.2 Exploiting $R^2$-sortability

We introduce '$R^2$-SortnRegress', an ordering-based search algorithm (Teyssier and Koller 2005) that exploits high $R^2$-sortability. This procedure follows *Var-SortnRegress*[2], a baseline algorithm that obtains state-of-the-art performance in causal discovery benchmarks with high var-sortability (Reisach et al. 2021). *Var-SortnRegress* is computationally similar to the *Bottom-Up* and *Top-Down* algorithms by Ghoshal and Honorio (2018) and Chen et al. (2019), which rely on an equal noise variance assumption. However, unlike the assumptions underlying these algorithms, the $R^2$ criterion is scale-invariant such that $R^2$-SortnRegress is unaffected by changes in data scale.

In $R^2$-SortnRegress (Algorithm 1), the coefficient of determination is estimated for every variable to obtain a candidate causal order. Then, a regression of each node onto its predecessors in that order is performed to obtain the candidate causal graph. To encourage sparsity, one can use a penalty function

---

[2]Reisach et al. (2021) refer to *Var-SortnRegress* as *SortnRegress*. We add the *Var* to emphasize the contrast to $R^2$-SortnRegress.

$\lambda(\theta)$. In our implementation, we use an L1 penalty with the regularization parameter chosen via the Bayesian Information Criterion (Schwarz 1978).

As before, we denote by $M_{t,S}^{\theta}$ a parameterized model of $X_t$ with covariates $X_S$ and parameters $\theta$. For an example, consider the linear model $M_{t,S}^{\theta} \colon \mathbf{X}_S \mapsto \langle \theta, \mathbf{X}_S \rangle$ with $\theta_S = \mathbb{R}^{|S|}$. To simplify notation, we define the mean squared error (MSE) of a model as $\text{MSE}(\mathbf{X}_t, M_{t,S}^{\theta}) = n^{-1} \sum_{i=1}^{n} (\mathbf{X}_t^{(i)} - M_{t,S}^{\theta}(\mathbf{X}_S^{(i)}))^2$.

---

**Algorithm 1:** $R^2$-SortnRegress

---

**Data:** $\mathbf{X} \in \mathbb{R}^{n \times d}$
**Input:** $\lambda$ /* Penalty function, for example $\lambda(\theta) = \|\theta\|_1$
**Result:** Weight matrix estimate $\widehat{W} \in \mathbb{R}^{d \times d}$
/* Candidate causal order
$\pi = \mathbf{0} \in \mathbb{R}^d$
/* Estimate $R^2$-s using $d$ regressions (with intercept; cf. Section 2)
**for** $t = 1, \ldots, d$ **do**
    $\theta^* = \arg\min_\theta \text{MSE}\left(\mathbf{X}_t, M_{t,\{1,\ldots,d\}\setminus\{t\}}^{\theta}\right)$
    $\pi_t = R^2(M_{t,\{1,\ldots,d\}\setminus\{t\}}^{\theta^*}, \mathbf{X})$
Find a permutation $\rho$ that sorts $\pi$ ascending
/* Estimate weights using $d - 1$ regressions (with intercept)
$\widehat{W} = \mathbf{0} \in \mathbb{R}^{d \times d}$
**for** $i = 2, \ldots, d$ **do**
    $t = \{j \colon \rho(j) = i\}$
    $S = \{j \colon \rho(j) < i\}$
    $\widehat{W}_{S,t} = \arg\min_\theta \text{MSE}\left(\mathbf{X}_t, M_{t,S}^{\theta}\right) + \lambda(\theta)$
**return** $\widehat{W}$

---

# 4 Utilizing $R^2$-Sortability for Causal Discovery

We compare $R^2$-*SortnRegress* to a var-sortability baseline and established structure learning algorithms in the literature across simulated data with varying $R^2$-sortabilities, and evaluate the algorithms on the real-world protein signalling dataset by Sachs et al. (2005).

## 4.1 Experiments on Simulated Data

We evaluate and compare $R^2$-*SortnRegress* on observations obtained from ANMs simulated with the following parameters:

$$P_{\mathcal{G}} \in \{\mathcal{G}_{\text{ER}}(20, 40), \mathcal{G}_{\text{SF}}(20, 40)\} \qquad \text{(Erdős–Rényi and scale-free graphs, } d = 20, \gamma = 2)$$
$$\mathcal{P}_N(\phi) = \mathcal{N}(0, \phi^2) \qquad \text{(Gaussian noise distributions)}$$
$$P_\sigma = \text{Unif}(0.5, 2) \qquad \text{(noise standard deviations)}$$
$$P_W = \text{Unif}((-1, -0.5) \cup (0.5, 1)) \qquad \text{(edge weights)}$$

In this Gaussian setting, the graph is not identifiable unless additional specific assumptions on the choice of edge and noise parameters are employed. We sample 500 ANMs (graph, noise standard deviations, and edge weights) using ER graphs, and another 500 using SF graphs. From each ANM, we generate 1000 observations. Data are standardized to mimic real-world settings with unknown data scales. We compare $R^2$-*SortnRegress* to the *PC* algorithm (Spirtes and Glymour 1991) and *FGES* (Meek 1997; Chickering 2002). Representative for scale-sensitive algorithms, we include *Var-SortnRegress*, introduced by Reisach et al. (2021) to exploit var-sortability, as well as their *RandomRegress* baseline (we refer to their article for comparisons to further algorithms). For details on the implementations used, see Appendix A. The completed partially directed acyclic graphs (CPDAGs) returned by *PC* and *FGES* are favorably transformed into DAGs by assuming the correct direction for any undirected edge if there is a corresponding directed edge in the true graph, and breaking any remaining cycles. We evaluate the structural intervention distance (SID) (Peters and Bühlmann 2015) between the recovered and the ground-truth DAGs. The SID counts the number of incorrectly estimated interventional distributions between node pairs when using parent-adjustment

in the recovered graph compared to the true graph (lower values are better). It can be understood as a measure of disagreement between a causal order induced by the recovered graph and a causal order induced by the true graph.

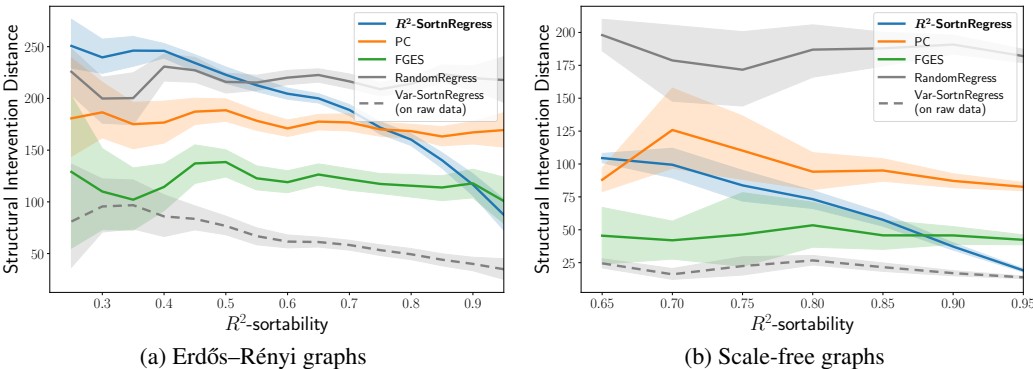

(a) Erdős–Rényi graphs                                (b) Scale-free graphs

Figure 1: Performance comparison on standardized synthetic data in terms of SID (lower is better) using moving window averages. $R^2$-*SortnRegress* (blue) performs well if $R^2$-sortability is high, achieving results competitive with established methods. For reference, we show the performance achieved by *Var-SortnRegress* on raw data (gray dashed line), which worsens to that of *Random-Regress* (gray) when the raw data scale is unavailable, as is the case here after standardization.

The results are shown in Figure 1. To analyze trends, we use window averaging with overlapping windows of a width of $0.1$ centered around $0.05, 0.1, \ldots, 0.95$. The lines indicate the change in mean SID from window to window, and the shaded areas indicate the 95% confidence interval of the mean. The means of non-empty windows start at $0.25$ for ER graphs and at $0.65$ for SF graphs. For both graph types, there are many instances with a $R^2$-sortability well above $0.5$. This is notable since we simulate a standard Gaussian non-equal variance setting with parameters chosen in line with common practice in the literature (see for example the default settings of the simulation procedures provided by Squires 2018; Kalainathan et al. 2020; Zhang et al. 2021). We present an analogous simulation with lower absolute edge weights in $(-0.5, 0.1) \cup (0.1, 0.5)$ in Appendix A.1, show results in terms of the structural Hamming distance (SHD) in Appendix A.2, and provide a visualization of the distribution of $R^2$-sortability for higher and lower edge weights in Appendix A.3.

$R^2$-*SortnRegress* successfully exploits high $R^2$-sortability for causal discovery. On ER graphs (Figure 1a), $R^2$-*SortnRegress* outperforms the *RandomRegress* baseline when $R^2$-sortability is greater than $0.5$. For $R^2$-sortabilities close to 1, $R^2$-*SortnRegress* outperforms *PC* and matches or exceeds the performance of *FGES*. On SF graphs (Figure 1b), $R^2$-sortabilities are generally higher, and $R^2$-*SortnRegress* outperforms *PC* and *FGES* for all but the lowest values, even coming close to the performance of *Var-SortnRegress* on raw data for very high $R^2$-sortabilities. On raw synthetic data, *Var-SortnRegress* successfully exploits the high var-sortability typical for many ANM simulation settings. Its performance improves for high $R^2$-sortability, showing the link between $R^2$-sortability and var-sortability. However, this performance relies on knowledge of the true data scale to exploit var-sortability. The same is true for other scale-sensitive scores such as the one used by *NOTEARS* (Zheng et al. 2018) and many other algorithms that build upon it. Without information in the variances (as is the case after standardization), *Var-SortnRegress* is reduced to *RandomRegress*, which consistently performs poorly across all settings. *FGES* outperforms *PC*, which may be explained by *FGES* benefiting from optimizing a likelihood specific to the setting, thus using information that is not available to *PC*. Further causal discovery results for settings with different noise and noise standard deviation distributions are shown in Appendix A.4.

## 4.2 Real-World Data

We analyze the observational part of the Sachs et al. (2005) protein signalling dataset to provide an example of what $R^2$-sortabilities and corresponding causal discovery performances may be expected in the real world. The dataset consists of $853$ observations of $11$ variables, connected by $17$ edges in the ground-truth consensus graph. We find the $R^2$-sortability to be $0.82$, which is above the neutral value of $0.5$, but not as high as the values for which $R^2$-*SortnRegress* outperforms the other methods

in the simulation shown in Section 4.1. We run the same algorithms as before on this data and evaluate them in terms of SID and SHD to the ground-truth consensus graph. To gain insight into the robustness of the results, we resample the data with replacement and obtain 30 datasets with a mean $R^2$-sortability of 0.79 and mean var-sortability of 0.65. The results (mean, [min, max]) are shown in the table below. We find that none of the algorithms perform particularly well (for reference: the empty graph has an SID of 53 and an SHD of 17). Although the other algorithms perform better than *RandomRegress* on average with *PC* performing best, the differences between them are less pronounced than in our simulation (Section 4.1).

| Algorithm | SID | SHD |
|---|---|---|
| $R^2$-SortnRegress | 48.13  [43, 51] | 14.43  [12, 17] |
| PC | 43.70  [37, 49] | 11.40  [ 9, 16] |
| FGES | 48.43  [45, 55] | 12.77  [11, 15] |
| RandomRegress | 52.57  [42, 59] | 16.20  [14, 19] |
| Var-SortnRegress | 45.13  [43, 47] | 15.53  [13, 18] |

These results showcase the difficulty of translating simulation performances to real-world settings. For benchmarks to be representative of what to expect in real-world data, benchmarking studies should differentiate between different settings known to affect causal discovery performance. This includes $R^2$-sortability, and we therefore recommend that benchmarks differentiate between data with different levels of $R^2$-sortability and report the performance of $R^2$-*SortnRegress* as a baseline.

# 5    Emergence and Prevalence of $R^2$-Sortability in Simulations

The $R^2$-sortability of an ANM depends on the relative magnitudes of the $R^2$ coefficients of determination of each variable given all others, and the $R^2$ coefficients in turn depend on the graph structure, noise variances, and edge weights. Although one can make the additional assumption of an equal fraction of cause-explained variance for all non-root nodes (see, for example, Sections 5.1 of Squires et al. 2022; Agrawal et al. 2023), this does not guarantee a $R^2$-sortability close to 0.5, and the $R^2$ may still carry information about the causal order as we show in Appendix E. Furthermore, one cannot isolate the effect of individual parameter choices on $R^2$-sortability, nor easily obtain the expected $R^2$-sortability when sampling ANMs given some parameter distributions, because the $R^2$ values are determined by a complex interplay between the different parameters. By analyzing the simpler case of causal chains, we show that the weight distribution plays a crucial role in the emergence of $R^2$-sortability. We also empirically assess $R^2$-sortability for different ANM simulations to elucidate the connection between $R^2$-sortability and ANM parameter choices.

## 5.1    The Edge Weight Distribution as a Driver of $R^2$-Sortability in Causal Chains

Our goal is to describe the emergence of high $R^2$ along a causal chain. We begin by analyzing the node variances. In a chain from $X_0$ to $X_p$ of a given length $p > 0$, the structure of $\mathcal{G}$ is fixed. Thus, sampling parameters consists of drawing edge weights $W_{0,1}, ..., W_{p-1,p} \sim P_W$ and noise standard deviations $\sigma_0, ..., \sigma_p \sim P_\sigma$ (cf. Section 2.1). Importantly, these parameters are commonly drawn iid, and we adopt this assumption here. As we show in Appendix B.1, the variance of the terminal node $X_p$ in the resulting chain ANM is lower bounded by the following function of the random parameters

$$\sigma_0^2 \sum_{j=0}^{p-1} \log |W_{j,j+1}|. \tag{3}$$

We assume that the distribution of noise standard deviations $P_\sigma$ has bounded positive support, as is commonly the case when sampling ANM parameters (see Squires 2018; Kalainathan et al. 2020; Zhang et al. 2021, as well as our simulations). We introduce the following sufficient condition for a weight distribution $P_W$ to result in diverging node variances along causal chains of increasing length

$$0 < \mathbb{E}\left[\log |V|\right] < +\infty, \text{ with } V \sim P_W. \tag{4}$$

If Equation (4) holds, the lower bound on the node variances given in Equation (3) diverges almost surely by the strong law of large numbers

$$\sigma_0^2 \sum_{j=0}^{p-1} \log |W_{j,j+1}| \xrightarrow[p \to \infty]{\text{a.s.}} +\infty,$$

since the $W_{j,j+1}$ are sampled iid from $P_W$. This implies that the variance of a terminal node in a chain of length $p$ also diverges almost surely to infinity as $p$ goes to infinity. Under the same condition, the $R^2$ of the terminal node given all other nodes converges almost surely to 1 as $p$ goes to infinity. We can show this using the fraction of cause-explained variance as a lower bound on the $R^2$ of a variable $X_p$ and the divergence of the variances to infinity:

$$R^2 = 1 - \frac{\text{Var}\left(X_p - \mathbb{E}\left[X_p \mid X_{\{0,\dots,d\}\setminus\{p\}}\right]\right)}{\text{Var}\left(X_p\right)} \geq 1 - \frac{\text{Var}\left(X_p - \mathbb{E}\left[X_p \mid X_{p-1}\right]\right)}{\text{Var}\left(X_p\right)}$$

$$= 1 - \frac{\sigma_p^2}{\text{Var}\left(X_p\right)} \xrightarrow[p\to\infty]{\text{a.s.}} 1.$$

For a $P_W$ with finite $\mathbb{E}\left[\log|V|\right]$ with $V \sim P_W$, it is thus sufficient that $\mathbb{E}\left[\log|V|\right] > 0$ for the terminal node's $R^2$ in a $p$-chain to converge to 1 almost surely as $p \to +\infty$, under iid sampling of edge weights and noise standard deviations. The fact that Equation (4) only depends on $P_W$ highlights the importance of the weight distribution as a driver of $R^2$.

**Increasingly Deterministic Relationships Along the Causal Order.** Our argument for the almost sure convergence of $R^2$ along causal chains relies on a) the divergence of the total variance of the terminal node, given Equation (4) holds, and b) independently drawn noise variances $\sigma^2$ from a bounded distribution. Together, these two observations imply that the relative contributions of the noise variances to the total variances tend to decrease along a causal chain. If data are standardized to avoid increasing variances (and high var-sortability), the noise variances tend to vanish[3] along the causal chain, resulting in increasingly deterministic relationships that effectively no longer follow an ANM.

## 5.2 Impact of the Weight Distribution on $R^2$-Sortability in Random Graphs

Understanding the drivers of $R^2$-sortability in random graphs is necessary to control the level of $R^2$-sortability in data simulated from ANMs with randomly sampled parameters. Given the importance of the weight distribution in our analysis of $R^2$-sortability in causal chains, we empirically evaluate the impact of the weight distribution on $R^2$-sortability in random graphs. We provide additional experiments showing how the weight distribution and average in-degree impact $R^2$-sortability in Appendix B.2.

We analyze $R^2$-sortability for different graph sizes. We use ER and SF graphs, that is $P_{\mathcal{G}} \in \{\mathcal{G}_{\text{ER}}(d, 2d), \mathcal{G}_{\text{SF}}(d, 2d)\}$, with $d$ nodes and $\gamma = 2$, and simulate graphs with $d \geq 5$ such that $2d$ does not exceed the maximum number of possible edges. Let $V$ be a random variable with $V \sim P_W$. We test different edge weight distributions $P_W = \text{Unif}((-\alpha, -0.1) \cup (0.1, \alpha))$ with $\alpha > 0.1$ chosen such that the $\mathbb{E}\left[\ln|V|\right]$ are approximately evenly spaced in $[-1, 2.5]$. We otherwise use the same settings as in Section 4.1. For each graph model, weight distribution, and number of nodes, we independently simulate 50 ANMs (graph, noise standard deviations, and edge weights).

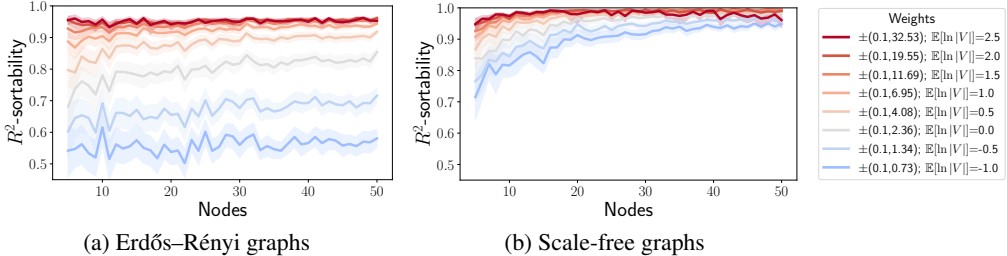

(a) Erdős–Rényi graphs          (b) Scale-free graphs

Figure 2: $R^2$-sortability at different graph sizes for a range of weight distributions with different $\mathbb{E}\left[\ln|V|\right]$ with $V \sim P_W$.

The results are shown in Figure 2. We observe that $R^2$-sortability is consistently higher for edge weight distributions $P_W$ with higher $\mathbb{E}\left[\ln|V|\right]$. For each weight distribution, we observe a slowing

---

[3]The noise contribution to a standardized variable $X_j^{\text{std}} = (w_{j-1,j}X_{j-1} + N_j)/\sqrt{\text{Var}\left(X_j\right)}$ in a causal chain, given as $\text{Var}(N_j/\sqrt{\text{Var}\left(X_j\right)})$, vanishes for $\text{Var}\left(X_j\right) \to +\infty$.

increase in $R^2$-sortability as graph size increases. Compared to ER graphs (Figure 2a), $R^2$-sortability in SF graphs tends to increase faster in graph size and is consistently higher on average (Figure 2b). In all cases, mean $R^2$-sortability exceeds $0.5$, and weight distributions with $\mathbb{E}\left[\ln|V|\right] > 0$ reach levels of $R^2$-sortability for which $R^2$-*SortnRegress* achieves competitive causal discovery performance in our simulation shown in Section 4.1. Settings with $\mathbb{E}\left[\log|V|\right] < 0$ may result in lower $R^2$-sortabilities, but also make detecting statistical dependencies between variables connected by long paths difficult as the weight product goes to $0$. In summary, we find that the weight distribution strongly affects $R^2$-sortability and can be used to steer $R^2$-sortability in simulations.

How the weight distribution affects $R^2$-sortability also depends on the density and topology of the graph, giving rise to systematic differences between ER and SF graphs. In Appendix B.2 we show the $R^2$-sortability for graphs of $50$ nodes with different $\mathbb{E}\left[\ln|V|\right]$ and average in-degree $\gamma$. In Erdős–Rényi graphs, we find that $R^2$-sortabilities are highest when neither of the parameters take extreme values, as is often the case in simulations. In scale-free graphs we observe $R^2$-sortabilities close to $1$ across all parameter settings. These results show that the graph connectivity interacts with the weight distribution and both can affect $R^2$-sortability and thereby the difficulty of the causal discovery task.

## 6 Discussion and Conclusion

We introduce $R^2$ as a simple scale-invariant sorting criterion that helps find the causal order in linear ANMs. Exploiting $R^2$-sortability can give competitive structure learning results if $R^2$-sortability is sufficiently high, which we find to be the case for many ANMs with iid sampled parameters. This shows that the parameter choices necessary for ANM simulations strongly affect causal discovery beyond the well-understood distinction between identifiable and non-identifiable settings due to properties of the noise distribution (such as equal noise variance or non-Gaussianity). High $R^2$-sortability is particularly problematic because high $R^2$ are closely related to high variances. When data are standardized to remove information in the variances, the relative noise contributions can become very small, resulting in a nearly deterministic model. Alternative ANM sampling procedures and the $R^2$-sortability of nonlinear ANMs and structural equation models without additive noise are thus of interest for future research. Additionally, determining what $R^2$-sortabilities may be expected in real-world data is an open problem. A complete theoretical characterization of the conditions sufficient and/or necessary for extreme $R^2$-sortability could help decide when to assume and exploit it in practice. Exploiting $R^2$-sortability could provide a way of using domain knowledge about parameters such as the weight distribution and the connectivity of the graph for causal discovery. On synthetic data, the prevalence of high $R^2$-sortability raises the question of how well other causal discovery algorithms perform relative to the $R^2$-sortability of the benchmark, and whether they may also be affected by $R^2$-sortability. For example, the same causal discovery performance may be less impressive if the $R^2$-sortability on a dataset is $0.95$ rather than $0.5$. In this context, the $R^2$-*SortnRegress* algorithm offers a strong baseline that is indicative of the difficulty of the causal discovery task. Our empirical results on the impact of the weight distribution and graph connectivity provide insight on how to steer $R^2$-sortability within existing linear ANM simulations. Simulating data with different $R^2$-sortabilities can help provide context to structure learning performance and clarifies the impact of simulation choices, which is necessary to evaluate the real-world applicability of causal discovery algorithms.

### Acknowledgements

We thank Brice Hannebicque for helpful discussions on an earlier draft of this paper. AGR received funding from the European Union's Horizon 2020 research and innovation programme under the Marie Skłodowska-Curie grant agreement No 945332 .

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

# Appendix

## A Causal Discovery Experiments

We provide an open-source implementation of $R^2$-sortability, $R^2$-*SortnRegress*, and an ANM sampling procedure in our library CausalDisco. In our experiments we make extensive use of the *NumPy* (Harris et al. 2020), *scikit-learn* (Pedregosa et al. 2011), and *statsmodels* (Seabold and Perktold 2010) Python packages, as well as the Python interface to the *igraph* (Csardi, Nepusz, et al. 2006) package. For the *PC* and *FGES* algorithms we use the implementation by Ramsey et al. (2018). The CPDAGs returned by *PC* and *FGES* are favorably transformed into DAGs by assuming the correct direction for any undirected edge if there is a corresponding directed edge in the true graph. To break any remaining cycles, we iteratively pick one of the shortest ones and remove one of the edges that break the most cycles until no cycles remain. For var-sortability and *Var-SortnRegress* we use the implementations provided by Reisach et al. (2021). In the experiments shown in Appendix A.4 we additionally show the *Top-Down* algorithm by Chen et al. (2019) using their implementation. To evaluate structure learning performance in SID (Peters and Bühlmann 2015) we run the authors' implementation using version 3.6.3 of the R programming language.

### A.1 Causal Discovery Performance and $R^2$-Sortability – Low Absolute Edge Weights

Given the impact of high absolute edge weights on $R^2$ as shown in Section 5, it may seem tempting to simulate ANMs with low absolute edge weights to avoid high levels of $R^2$-sortability. To assess the potential of this idea, we simulate a setting with low absolute edge weights. The simulation in Section 4.1 draws from $P_W = \text{Unif}((-2, -0.5) \cup (0.5, 2))$, resulting in $\mathbb{E}\left[\log|V|\right] \approx 0.16$ with $V \sim P_W$. Here, we draw from $P_W = \text{Unif}((-0.5, -0.1) \cup (0.1, 0.5))$, resulting in $\mathbb{E}\left[\log|V|\right] \approx -1.29$ with $V \sim P_W$, while keeping the remaining settings the same as in Section 4.1.

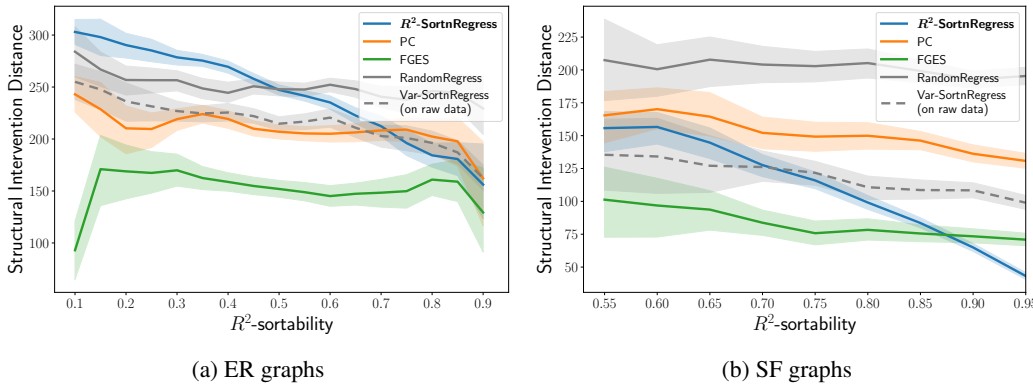

(a) ER graphs                                               (b) SF graphs

Figure 3: Performance comparison on simulated data with low absolute edge weights in terms of SID (lower is better). $R^2$-*SortnRegress* (blue) performs well if $R^2$-sortability is high, achieving results competitive with established methods. For reference, we show the performance achieved by *Var-SortnRegress* on raw data (gray dashed line), which worsens to that of *RandomRegress* (gray) when the raw data scale is unavailable, as is the case here after standardization.

The results are shown in Figure 3. Compared to the setting with higher absolute edge weights shown in Section 4.1, we observe lower values of $R^2$-sortability overall. On ER graphs, as shown in Figure 3a, $R^2$-*SortnRegress* performs worse than *RandomRegress* for $R^2$-sortabilities below 0.5 and outperforms it for $R^2$-sortabilities greater than 0.5, as expected from the definition of $R^2$-sortability which yields 0.5 to be the neutral value. For $R^2$-sortabilities closer to 1, $R^2$-*SortnRegress* outperforms *PC* and narrows the gap to *FGES*. *Var-SortnRegress* performs similar to *PC* and worse than *FGES* in this setting, likely because the lower edge weights mitigates the variance explosion that characterizes settings with higher edge weights (Reisach et al. 2021). On SF graphs, as shown in Figure 3b, $R^2$-sortabilities are generally above the neutral value of 0.5 despite the lower absolute edge weights, and $R^2$-*SortnRegress* performs well, even outperforming all other algorithms on many instances. The

excellent absolute and relative performance of $R^2$-*SortnRegress* and the high values of $R^2$-sortability indicate that the SF graph structure may be particularly favorable for an exploitable $R^2$ pattern to arise. Choosing lower absolute edge weights results in lower $R^2$-sortabilities and degrades the comparative performance of $R^2$-*SortnRegress* on ER graphs and, to a lesser extent, also on SF graphs. However, $R^2$-*SortnRegress* still performs excellently on SF graphs and can achieve good results on ER graphs with high $R^2$-sortability, which do occur despite the low absolute edge weights. Overall, extreme $R^2$-sortabilities are less frequent for low absolute edge weights, but they are not impossible and can impact causal discovery performance. We therefore recommend reporting $R^2$-sortability regardless of the edge weight range.

## A.2 Evaluation of Causal Discovery Results Using Structural Hamming Distance

Here we show causal discovery performances in terms of the structural Hamming distance (SHD). Note that SHD, although frequently used as a performance measure in causal discovery, cannot be directly interpreted in terms of the suitability of the discovered graph for effect estimation, except if it is zero. For example, excess edges to causal predecessors are treated the same as excess edges to causal successors in the true graph despite their different implications for effect estimation. For the same reason, SHD results are very sensitive to the density of the true graph. As a result, the absolute performance of regression-based methods such as $R^2$-*SortnRegress*, *Var-SortnRegress*, and *RandomRegress* strongly depends on the the sparsity penalty used. For additional context, recall that the empty graph has a constant SHD in the product of average in-degree and number of nodes $\gamma d$, which is equal to $40$ for the settings shown here.

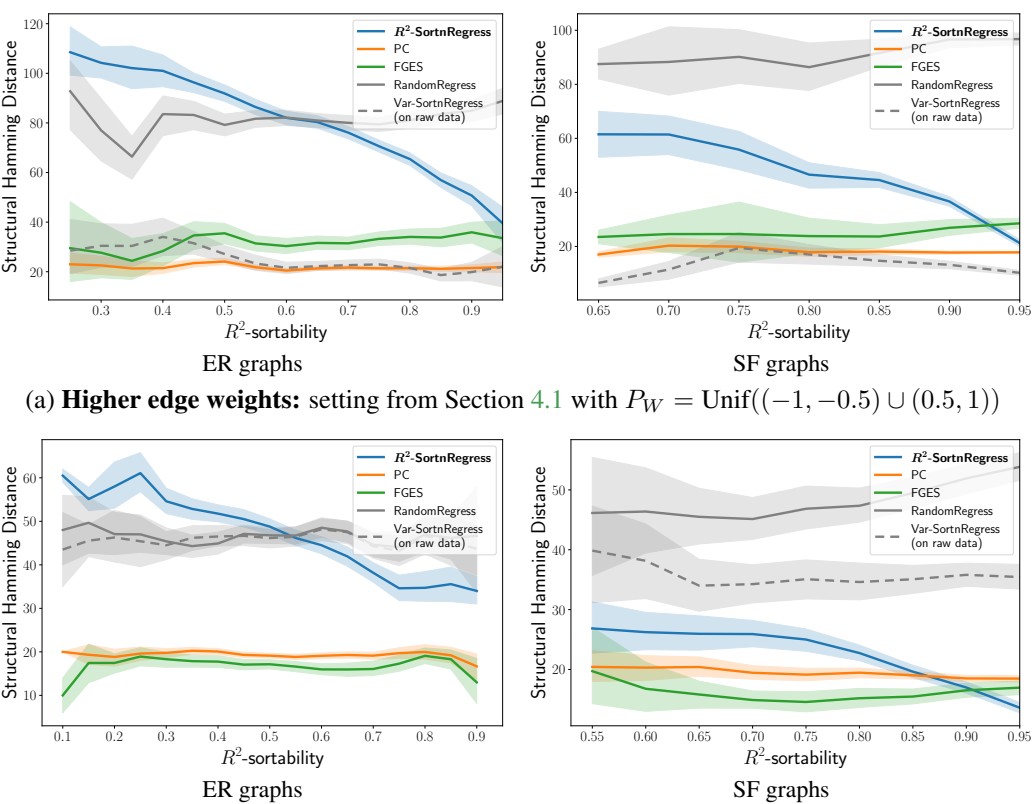

(a) **Higher edge weights:** setting from Section 4.1 with $P_W = \text{Unif}((-1, -0.5) \cup (0.5, 1))$

(b) **Lower edge weights:** setting from Appendix A.1 with $P_W = \text{Unif}((-0.5, -0.1) \cup (0.1, 0.5))$

Figure 4: Performance comparison on simulated data with high and low absolute edge weights in terms of SHD (lower is better). $R^2$-*SortnRegress* (blue) performance improves with higher $R^2$-sortability, in a trend similar to that observed for SID (Section 4.1 and Appendix A.1).

As can be seen in Figure 4, the trends in performance for $R^2$-*SortnRegress* are the same as in terms of SID (Section 4.1 and Appendix A.1). The fact that the results of $R^2$-*SortnRegress* are comparatively

better in terms of SID than in terms of SHD indicates that the algorithm finds a good causal order and inserts excess edges that do not affect the correctness of adjustment sets as measured by SID. The same logic applies to *FGES*, which consistently performs better than *PC* in terms of SID (Section 4.1 and Appendix A.1) but does not perform better consistently in terms of SHD (Figure 4).

### A.3 Distribution of $R^2$-Sortability

In Figure 5 we provide the $R^2$-sortability histograms for the settings described in Section 4 and Appendix A.1 as reference points for the distribution of $R^2$-sortability in random graphs sampled with common settings. We observe that the distribution of $R^2$-sortability is close to symmetric in ER graphs, while for SF graphs it has a strong left skew. The mean values are higher when edge weights are high (Figure 5a) than when they are low (Figure 5b), as is expected from our reasoning about the impact of the weight distribution outlined in Section 5.

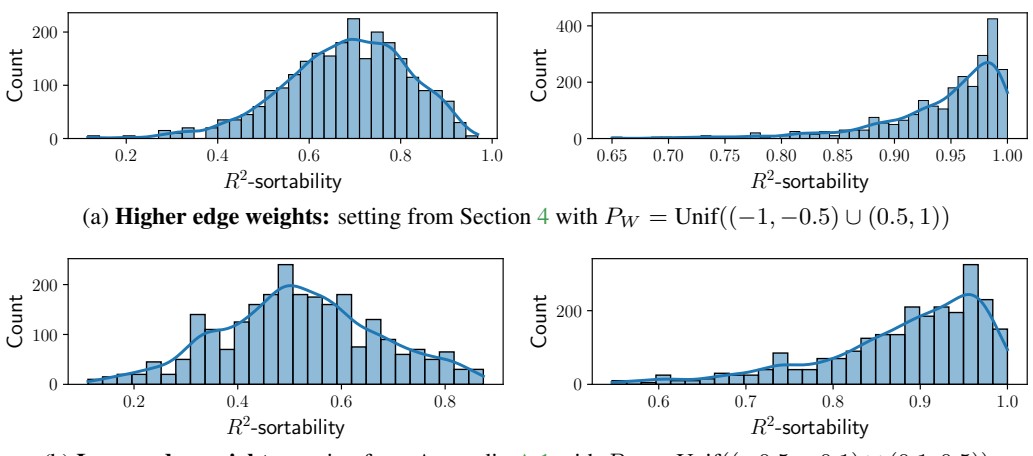

(a) **Higher edge weights:** setting from Section 4 with $P_W = \text{Unif}((-1, -0.5) \cup (0.5, 1))$

(b) **Lower edge weights:** setting from Appendix A.1 with $P_W = \text{Unif}((-0.5, -0.1) \cup (0.1, 0.5))$

Figure 5: Histograms and kernel density estimates of the distribution of $R^2$-sortability in different simulations.

### A.4 Causal Discovery Results For Different Noise Settings

Many identifiability results in the ANM literature focus on the distribution of the additive noise and its variance, and the distinction between different distributions is common in the evaluation of causal discovery algorithms. Here, we present $R^2$-*SortnRegress* results and $R^2$-sortability separately for different noise distributions and variances and show results in terms of SID and SHD (note that $R^2$-*SortnRegress* is regularized using the Bayesian Information Criterion, which effectively strikes a balance between SID and SHD). We also provide an evaluation of the performance in terms of the discovery of the Markov Equivalence Class (MEC). We transform recovered DAGs into corresponding CPDAGs before calculating the MEC-SID, and additionally transform true DAGs into corresponding CPDAGs before evaluating the MEC-SHD. For each setting we independently sample 30 graphs using the edge weight distribution $P_W = \text{Unif}((-2, -0.5) \cup (0.5, 2))$ and run our algorithms on 1000 iid observations from each ANM.

The causal discovery performances and $R^2$-sortabilities of each setting are shown in Figure 6. We find that $R^2$-sortability is consistently high on average ($\approx 0.8$), except for the setting with exponentially distributed noise standard deviations ($\approx 0.65$) shown in Figure 6b. Unlike the uniform distribution used in the other settings, the exponential distribution is positively unbounded, meaning that there is always a chance that the noise standard deviations are large enough to substantially impact $R^2$-sortability. In the settings with high average $R^2$-sortability, $R^2$-*SortnRegress* performs similarly to *FGES* in terms of SID, and somewhat worse in terms of SHD. In Figure 6b, the setting with exponentially distributed $\sigma$ and average $\mathbf{v}_{R^2} \approx 0.65$, $R^2$-*SortnRegress* expectedly performs worse than in the other settings. On raw data, the *Top-Down* algorithm performs well when the equal noise variance assumption is approximately fulfilled but poorly otherwise (e.g. Figure 6c), and consistently performs similarly or worse than *RandomRegress* on standardized data. The performance of *PC* and

*FGES* is consistent across DAG recovery settings, with only a slight dip for *FGES* in the case of non-Gaussian noise shown in Figure 6c where its likelihood is misspecified. Conversely, *DirectLiNGAM* performs well in the setting with non-Gaussian noise, but poorly in the other settings. We find qualitatively similar results for MEC recovery shown in Figure 6d, although the *RandomRegress* baseline appears to be somewhat stronger, likely because the direction of edges is less impactful for MEC comparisons.

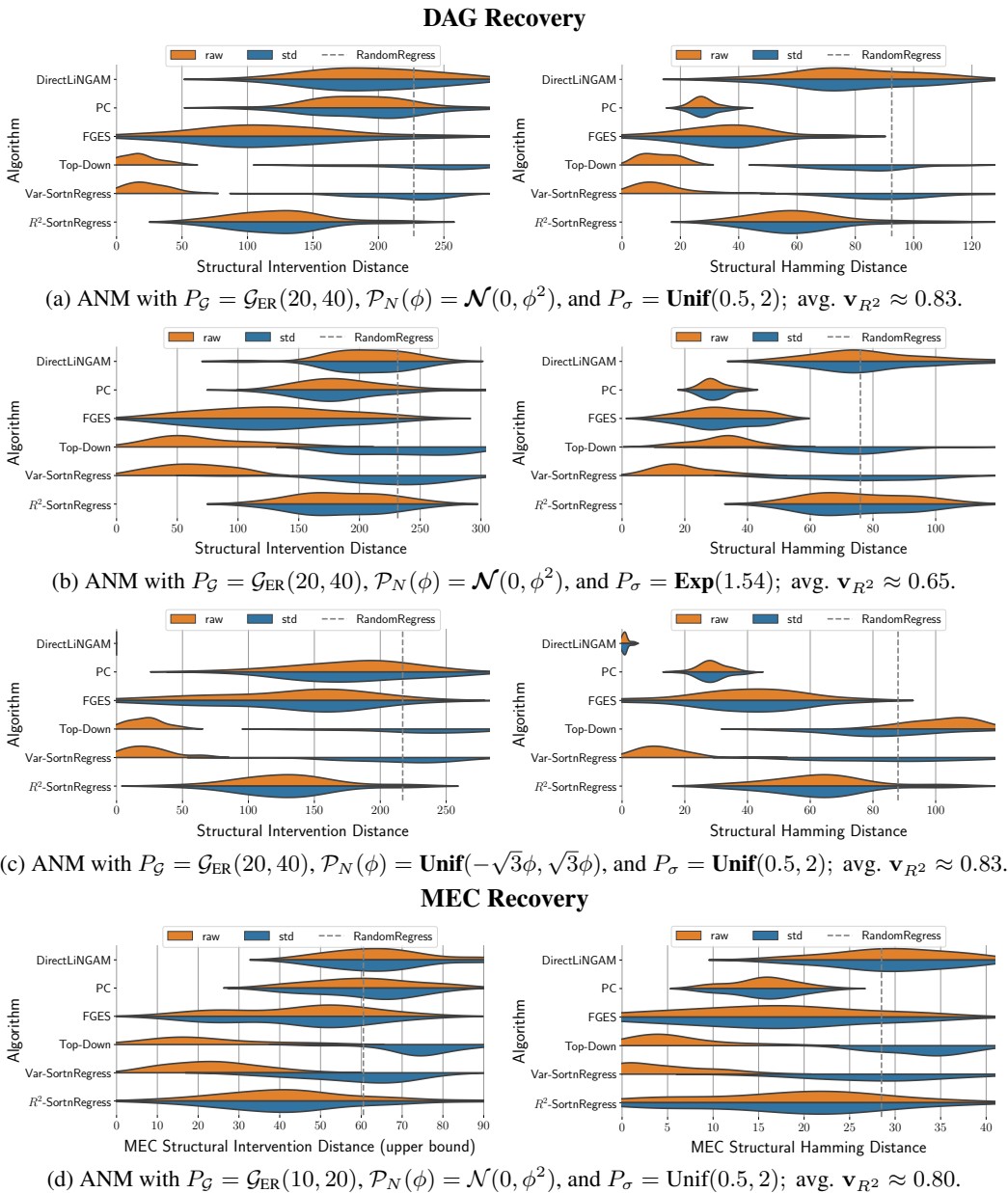

(a) ANM with $P_{\mathcal{G}} = \mathcal{G}_{\mathrm{ER}}(20, 40)$, $\mathcal{P}_N(\phi) = \mathcal{N}(0, \phi^2)$, and $P_\sigma = \mathbf{Unif}(0.5, 2)$; avg. $\mathbf{v}_{R^2} \approx 0.83$.

(b) ANM with $P_{\mathcal{G}} = \mathcal{G}_{\mathrm{ER}}(20, 40)$, $\mathcal{P}_N(\phi) = \mathcal{N}(0, \phi^2)$, and $P_\sigma = \mathbf{Exp}(1.54)$; avg. $\mathbf{v}_{R^2} \approx 0.65$.

(c) ANM with $P_{\mathcal{G}} = \mathcal{G}_{\mathrm{ER}}(20, 40)$, $\mathcal{P}_N(\phi) = \mathbf{Unif}(-\sqrt{3}\phi, \sqrt{3}\phi)$, and $P_\sigma = \mathbf{Unif}(0.5, 2)$; avg. $\mathbf{v}_{R^2} \approx 0.83$.

(d) ANM with $P_{\mathcal{G}} = \mathcal{G}_{\mathrm{ER}}(10, 20)$, $\mathcal{P}_N(\phi) = \mathcal{N}(0, \phi^2)$, and $P_\sigma = \mathrm{Unif}(0.5, 2)$; avg. $\mathbf{v}_{R^2} \approx 0.80$.

Figure 6: Causal discovery performance and $R^2$-sortability for different $\mathcal{P}_N$ and $P_\sigma$.

# B $R^2$-sortability for Different Simulation Parameter Choices

## B.1 Terminal Node Variance in a Causal Chain Given Weights and Noise Standard Deviations

In a causal chain $(X_0 \xrightarrow{w_{0,1}} X_1 \xrightarrow{w_{1,2}} X_2 \xrightarrow{w_{2,3}} \cdots \xrightarrow{w_{p-1,p}} X_p)$ of length $p > 0$ with fixed edge weights $w_{j,j+1}$ for $j = 0, ..., p-1$ and standard deviations $\boldsymbol{\sigma}_j$ for $j = 0, ..., p$ of the independent

noise variables $N_0, ..., N_p$, we have that

$$\text{Var}\left(X_p\right) = \text{Var}\left(w_{p-1,p}X_{p-1}\right) + \text{Var}\left(N_p\right) = \cdots \text{(unfolding the recursion)} \cdots$$

$$= \sum_{i=0}^{p-1} \text{Var}\left(N_i\right) \left(\prod_{j=i}^{p-1} w_{j,j+1}\right)^2 + \text{Var}\left(N_p\right) = \sum_{i=0}^{p-1} \boldsymbol{\sigma}_i^2 \left(\prod_{j=i}^{p-1} w_{j,j+1}\right)^2 + \boldsymbol{\sigma}_p$$

$$\geq \boldsymbol{\sigma}_0^2 \prod_{j=0}^{p-1} w_{j,j+1}^2 \geq \boldsymbol{\sigma}_0^2 \sum_{j=0}^{p-1} \log|w_{j,j+1}|.$$

The application of the log in the final step allows us to lower-bound the product by a sum and employ the law of large numbers. In Figure 7 we present a simulation of the convergence of $R^2$ and the cause-explained variance fraction alongside the increase of variances along the causal order to illustrate our theoretical results given in Section 5.1. We sample edge weights $P_W \sim \text{Unif}((-2, -0.5), (0.5, 2))$ and noise standard deviations $P_\sigma \sim \text{Unif}(0.5, 2)$. We sample 30 chains independently in this way to obtain confidence intervals.

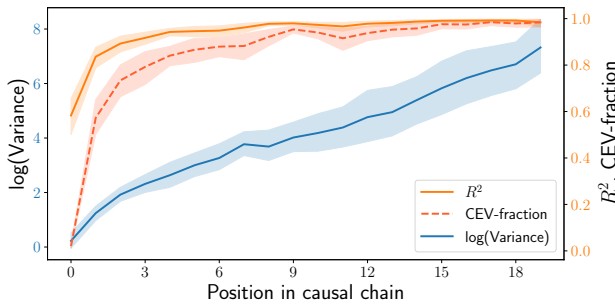

Figure 7: $R^2$, fraction of cause-explained variance (CEV), and variances in simulated causal chains.

## B.2   Empirical Evaluation of $R^2$-Sortability in ANMs

In this section we present the $R^2$-sortability obtained for graphs with a range of average node in-degrees $\gamma$ and weight distributions $P_W$ across random graph models, noise distributions, and noise standard deviation distributions. For each combination of parameters, we compute the mean value of $R^2$-sortability $\mathbf{v}_{R^2}$ for 20 independently sampled graphs with $d = 50$ nodes and 1000 iid observations per ANM.

### ER and SF Graph Models

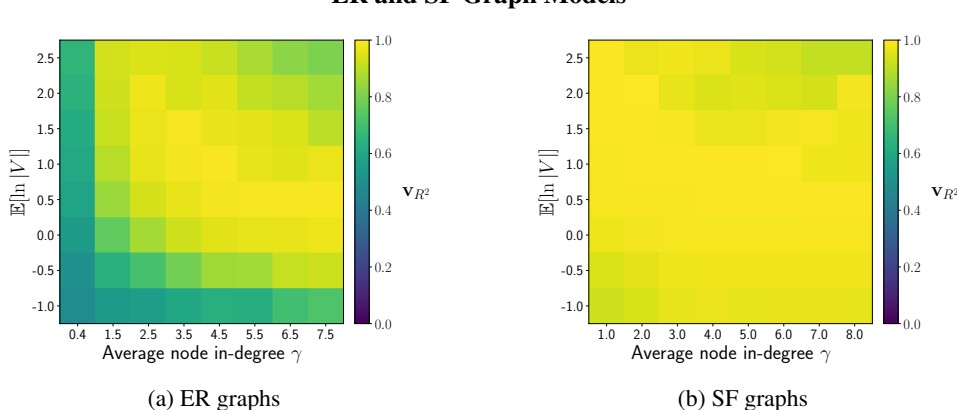

(a) ER graphs

(b) SF graphs

Figure 8: Sensitivity of $R^2$-sortability to parameters in 50-node graphs with $\mathcal{P}_N(\phi) = \mathcal{N}(0, \phi^2)$, $P_\sigma \sim \text{Unif}(0.5, 2)$ to $\gamma$ and $P_W$ with different $\mathbb{E}\left[\ln|V|\right]$ with $V \sim P_W$ in ER and SF random graphs.

**Different Noise Distributions and Noise Standard Deviations**

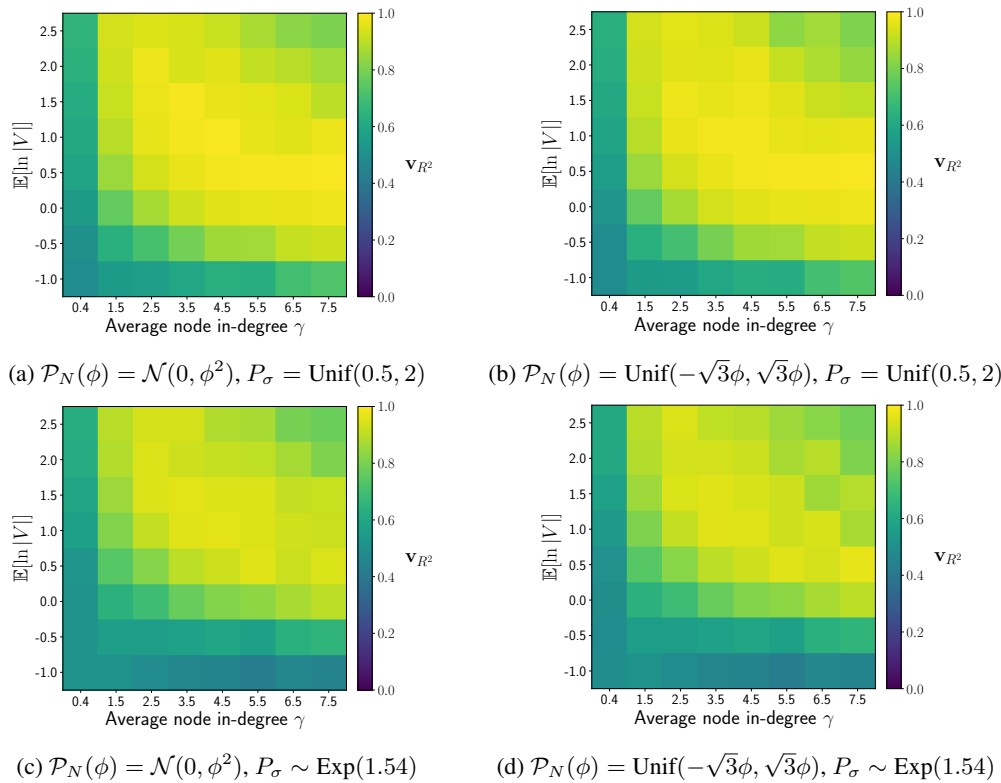

(a) $\mathcal{P}_N(\phi) = \mathcal{N}(0, \phi^2)$, $P_\sigma = \mathrm{Unif}(0.5, 2)$

(b) $\mathcal{P}_N(\phi) = \mathrm{Unif}(-\sqrt{3}\phi, \sqrt{3}\phi)$, $P_\sigma = \mathrm{Unif}(0.5, 2)$

(c) $\mathcal{P}_N(\phi) = \mathcal{N}(0, \phi^2)$, $P_\sigma \sim \mathrm{Exp}(1.54)$

(d) $\mathcal{P}_N(\phi) = \mathrm{Unif}(-\sqrt{3}\phi, \sqrt{3}\phi)$, $P_\sigma \sim \mathrm{Exp}(1.54)$

Figure 9: Sensitivity of $R^2$-sortability in $\mathcal{G}_{\mathrm{ER}}(50, \gamma 50)$ graphs to $\gamma$ and $P_W$ with different $\mathbb{E}[\ln|V|]$ with $V \sim P_W$, for different noise distributions $\mathcal{P}_N(\phi)$ and noise standard deviation distributions $P_\sigma$.

Figure 8 shows a comparison of $R^2$-sortability in ER and SF graphs. We choose slightly different values of $\gamma$ for the two settings because the SF graph generating mechanism requires integer values. In ER graphs, we observe that $R^2$-sortability is high unless $\mathbb{E}[\log|V|]$ or $\gamma$ are very low. In SF graphs we observe $R^2$-sortabilities close to 1 across all settings, the relationship between the parameters and $R^2$-sortability observed for ER graphs is only faintly visible.

Figure 9 shows the impact of the noise and noise standard deviation distributions on $R^2$-sortability. We observe a consistent trend of high $R^2$-sortability unless $\mathbb{E}[\log|V|]$ or $\gamma$ are very low. $R^2$-sortabilities also appear somewhat lower when edge weights and $\gamma$ are simultaneously very high. However, we find that high $\mathbb{E}[\log|V|]$ and $\gamma$ yield $R^2$ extremely close to 1 for many variables. What looks like lower $R^2$-sortability may therefore be due to finite samples and limited numerical precision. $R^2$-sortability does not differ visibly between different noise distributions $\mathcal{P}_N(\phi)$, but comparing the first to the second row, we observe that exponentially distributed noise standard deviations $\sigma$ leads to lower $R^2$-sortability, provided the $\mathbb{E}[\log|V|]$ are not too high. This observation is in line with the comparatively low $R^2$-sortability observed for exponentially distributed $\sigma$ in Appendix A.4. It may be explained by the positive unboundedness of the exponential distribution, which enables the noise variances to disrupt $R^2$ patterns even when cause-explained variance is high.

## C  Relationship Between the Different $\tau$-Sortabilities

Our use of $R^2$ is motivated by reasoning about an increase in the fraction of cause-explained variance (CEV) along the causal order for data with high var-sortability (see Section 3.1). We obtain CEV-sortability using the definition of $\tau$-sortability in Equation (2) with $\tau(X, t) = R^2(M_{t,\mathbf{Pa}(X_t)}^{\theta^*}, X)$, and denote it as $\mathbf{v}_{\mathrm{CEV}}$. Here, we analyze the relationship between var-sortability, $R^2$-sortability, and CEV-sortability to gain insight into the co-occurrence of var-sortability (which has been shown to be prevalent in many simulation settings, see Reisach et al. 2021) and $R^2$-sortability, as well as

the match between $R^2$-sortability and CEV-sortability.[4] For each combination of parameters, we compute the mean sortability of 20 independently sampled graphs with $d = 50$ nodes and 1000 iid observations per ANM.

## C.1 Erdős–Rényi Graphs

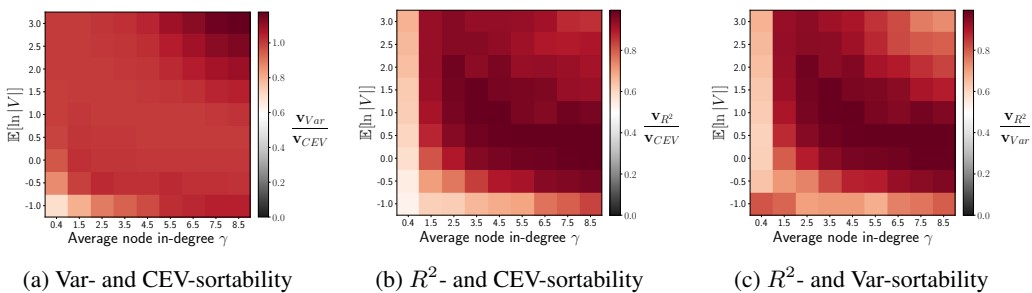

(a) Var- and CEV-sortability     (b) $R^2$- and CEV-sortability     (c) $R^2$- and Var-sortability

Figure 10: Ratio between sortabilities for $P_W$ with different $\mathbb{E}\left[\ln |V|\right]$ with $V \sim P_W$ and different $\gamma$ in $\mathcal{G}_{ER}(50, \gamma 50)$ graphs with $\mathcal{P}_N(\phi) = (0, \phi^2)$ and $P_\sigma = \text{Unif}(0.5, 2)$.

We can see in Figure 10a that CEV-sortability indeed tracks var-sortability closely in most settings as indicated by our analysis of the weight distribution in Section 5. The two measures disagree most when the parameters $\mathbb{E}\left[\log |V|\right]$ and $\gamma$ both take very low values. In Figure 10b, we see that $R^2$-sortability is generally a good proxy for CEV-sortability except when $\mathbb{E}\left[\log |V|\right]$ or $\gamma$ are very low. Figure 10c shows that $R^2$-sortability tracks var-sortability well across many settings, unless one of the parameters is very low. The disagreement for very high values of $\gamma$ and $\mathbb{E}\left[\log |V|\right]$ may be due to the difficulty of distinguishing between $R^2$ (or CEV-fractions) numerically in finite samples as they converge to 1.

## C.2 Scale-Free Graphs

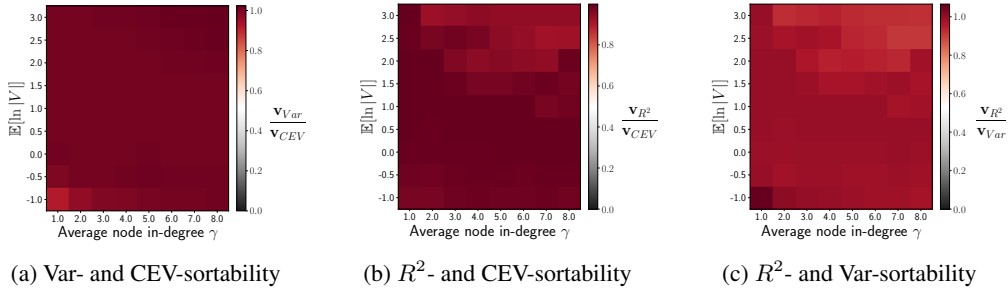

(a) Var- and CEV-sortability     (b) $R^2$- and CEV-sortability     (c) $R^2$- and Var-sortability

Figure 11: Ratio between sortabilities for $P_W$ with different $\mathbb{E}\left[\ln |V|\right]$ with $V \sim P_W$ and different $\gamma$ in $\mathcal{G}_{SF}(50, \gamma 50)$ graphs with $\mathcal{P}_N(\phi) = (0, \phi^2)$ and $P_\sigma = \text{Unif}(0.5, 2)$.

We choose slightly different values for $\gamma$ compared to Figure 10 because the scale-free graph generating mechanism requires integer values. Figure 11 shows that var-sortability, $R^2$-sortability, and CEV-sortability are almost perfectly aligned in all settings. The patterns visible in Figure 10 for ER graphs are only faintly visible for SF graphs, and the alignment between the measures is consistently close.

## D Definition of Sortability

The original definition of var-sortability by Reisach et al. (2021), given in Equation (2) for $\tau(X, t) = \text{Var}(X_t)$, measures the fraction of all directed paths of different length between cause-effect pairs

---

[4]Note that the range of values and the interpretation of the color scale differs between figures.

that are correctly ordered by variance. We investigate two alternative definitions of sortability to explore the impact of counting the directed paths between a cause-effect pair differently.

## D.1 Path-Existence (PE) Sortability

The first alternative sortability definition measures the fraction of directed cause-effect pairs (meaning that there exists at least one directed path from cause to effect) that are correctly sorted by a criterion $\tau$. Since this definition treats all connected pairs equally, irrespectively of the number or length of the connecting paths, we refer to it as 'Path-existence sortability'. It is given as

$$\mathbf{v}_\tau^{(\text{PE})}(X, \mathcal{G}) = \frac{\sum_{(s \to t) \in \sum_{i=1}^d B_\mathcal{G}^i} \text{incr}(\tau(X, s), \tau(X, t))}{\sum_{(s \to t) \in \sum_{i=1}^d B_\mathcal{G}^i} 1} \quad \text{where incr}(a, b) = \begin{cases} 1 & a < b \\ 1/2 & a = b \\ 0 & a > b \end{cases} \quad (5)$$

and $\sum_{i=1}^d B_\mathcal{G}^i$ is the sum of the matrices obtained by taking the binary adjacency matrix $B_\mathcal{G}$ of graph $\mathcal{G}$ to the $i$-th power. $(s \to t) \in \sum_{i=1}^d B_\mathcal{G}^i$ is true for $s, t$ if and only if there is at least one directed path from $X_s$ to $X_t$ in $\mathcal{G}$.

## D.2 Path-Count (PC) Sortability

The second alternative sortability definition measures the fraction of all directed paths between cause-effect pairs that are correctly sorted by a criterion $\tau$. This definition can be understood as weighing cause-effect pairs by the number of connecting paths, and we therefore refer to it as 'Path-count sortability'. It is given as

$$\mathbf{v}_\tau^{(\text{PC})}(X, \mathcal{G}) = \frac{\sum_{i=1}^d \sum_{(s \to t) \in B_\mathcal{G}^i} \left(B_\mathcal{G}^i\right)_{s,t} \text{incr}(\tau(X, s), \tau(X, t))}{\sum_{i=1}^d \sum_{(s \to t) \in B_\mathcal{G}^i} \left(B_\mathcal{G}^i\right)_{s,t}} \quad \text{where incr}(a, b) = \begin{cases} 1 & a < b \\ 1/2 & a = b \\ 0 & a > b \end{cases} \quad (6)$$

and $\left(B_\mathcal{G}^i\right)_{s,t}$ is the entry at position $s, t$ of the binary adjacency matrix $B_\mathcal{G}$ of graph $G$ taken to the $i$-th power, which is equal to the number of distinct directed paths of length $i$ from $X_s$ to $X_t$ in $\mathcal{G}$. $(s \to t) \in B_\mathcal{G}^i$ is true for $s, t$ if and only if there is at least one directed path from $X_s$ to $X_t$ of length $i$ in $\mathcal{G}$.

## D.3 Comparison of Sortability Definitions

We conduct an experiment to compare the definitions of sortability outlined above to the original definition presented in Equation (2) using $\tau(X, t) = \text{Var}(X_t)$ to obtain var-sortability and $\tau(X, t) = R^2(M_{\{1,\ldots,d\} \setminus \{t\}}^{\theta^*}, s)$ to obtain $R^2$-sortability, as in Section 3.1. We independently sample 500 ANMs with the parameters $P_\mathcal{G} = \mathcal{G}_{\text{ER}}(20, 40)$, $\mathcal{P}_N(\phi) = \mathcal{N}(0, \phi^2)$, $P_\sigma = \text{Unif}((-2, -0.5) \cup (0.5, 2))$, and $P_W = \text{Unif}((-0.5, -0.1) \cup (0.1, 0.5))$. From each ANM, we sample 1000 iid observations.

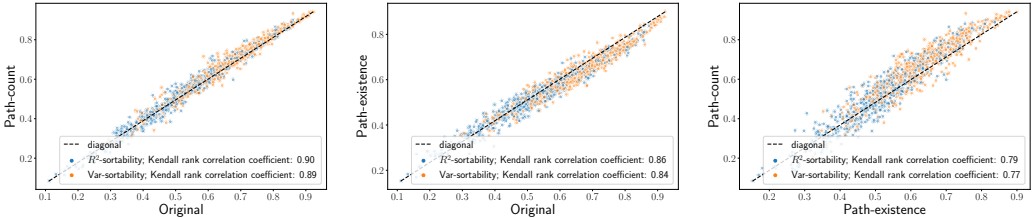

Figure 12: **Erdős–Rényi graphs:** comparison of the sortabilities obtained as defined by Equation (2) (following the original definition by Reisach et al. 2021) to alternative definitions that count directed paths between a cause-effect pair differently.

Figure 12 shows a comparison of the different definitions of sortability. We can see that the alternatives are highly correlated with the original definition for both $R^2$-sortability and var-sortability in terms of the Kendall rank correlation coefficient (Kendall 1938). The rank correlation coefficient between

path-existence and path-count sortability is somewhat lower, indicating that the original definition of sortability strikes a balance between the two alternatives. The point clouds are also close to the diagonal, meaning that the differences in sortability for individual graphs tend to be small. However, for graph structures with different connectivity patters, the differences between the definitions may be more pronounced. To investigate this aspect, we run another simulation in the same fashion, this time using 500 scale-free graphs ($P_{\mathcal{G}} = \mathcal{G}_{\text{SF}}(20, 40)$), and the same parameters otherwise.

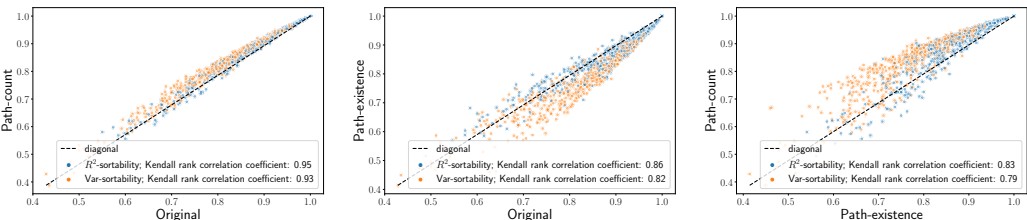

Figure 13: **Scale-free graphs:** comparison of the sortabilities obtained as defined by Equation (2) (following the original definition by Reisach et al. 2021) to alternative definitions that treat the connectivity of a cause-effect pair differently.

As can be seen in Figure 13, for scale-free graphs the original definition of sortability is similar to the path-count definition, and somewhat less so to the path-existence definition. As before, we observe the biggest difference in terms of rank correlation between path-count and path-existence sortability. While the rank correlations are close to those observed in Figure 12, there is a stronger trend of path-count sortability tending to be higher, and path-existence sortability tending to be lower than those of the original definition. In addition, the trends appear to be different for $R^2$-sortability and var-sortability, in particular when comparing path-existence sortability to the other two definitions. Overall, we find that the original definition strikes a balance between the alternatives. Nonetheless, the deviation from the diagonal and difference between $R^2$-sortability and var-sortability may indicate that some definitions could be more appropriate than others depending on the choice of $\tau$ and the assumptions about the graph structure.

## E   Controlling CEV Does Not Guarantee Moderate $R^2$-sortability

One strategy for avoiding var-sortability that may be interesting in the light of $R^2$-sortability is to enforce equal node variances by assuming constant cause-explained variance and constant exogenous noise variance, thereby also fixing the fraction of cause-explained variance (Squires et al. 2022; Agrawal et al. 2023, Sections 5.1).

In a causal chain, for the variances and exogenous noise fractions to be equal, the edge weights also have to be of equal magnitude. For this reason, in a such a chain we can at best hope to order the first and last node correctly using $R^2$, resulting in a $R^2$-sortability close to $\frac{1}{2}$ if the chain is sufficiently long.

In general however, the effect of such a sampling scheme on $R^2$-sortability is more complex. Using the parametrization described in Sections 5.1 of Squires et al. (2022), which enforces unit variances and a noise contribution of half of the total variance (except for root nodes, which have only exogenous variance), we provide a counterexample to show that controlling the fraction of cause-explained variance does not guarantee $R^2$-sortability close to $0.5$. Consider the causal model

$$X_1 = N_1$$
$$X_2 = \frac{X_1}{\sqrt{2}} + N_2$$
$$X_3 = \frac{X_1 + X_2}{\sqrt{4 + 2\sqrt{2}}} + N_3$$
$$X_4 = \frac{X_1 + X_2 + X_3}{\sqrt{6 + \frac{4}{\sqrt{2}} + 8\frac{1 + \frac{1}{\sqrt{2}}}{\sqrt{4 + 2\sqrt{2}}}}} + N_4$$

where all $N_i$ are independent and the root cause noise $N_1 \sim \mathcal{N}(0, 1)$, while $N_2, N_3, N_4 \sim \mathcal{N}(0, \frac{1}{2})$ such that all variables other than the root cause have a cause-explained variance fraction of $\frac{1}{2}$, and all variables have a marginal variance equal to 1, as in Section 5.1 of Squires et al. (2022). We are interested in the $R^2(X, t)$ for all $t \in \{1, 2, 3, 4\}$. We can use partial correlations to analytically compute

$$R^2(X, t) = 1 - \frac{1}{(\Sigma^{-1})_{t,t}},$$

where $\Sigma$ is the covariance matrix of $X$, which can be obtained analytically by computing the pairwise covariances using the definition of the random variables. Using the definitions

$$\alpha = \frac{1}{\sqrt{2}}, \quad \beta = \frac{1+\alpha}{\sqrt{4+2\sqrt{2}}}, \quad \gamma = \frac{(1+\alpha)\left(1 + \frac{1}{\sqrt{4+2\sqrt{2}}}\right)}{\sqrt{6+4\alpha+8\beta}}, \quad \delta = \frac{1 + \frac{2+\sqrt{2}}{\sqrt{4+2\sqrt{2}}}}{\sqrt{6+4\alpha+8\beta}}$$

we can write the covariance matrix of $(X_1, X_2, X_3, X_4)$ as

$$\Sigma = \begin{bmatrix} 1 & \alpha & \beta & \gamma \\ \alpha & 1 & \beta & \gamma \\ \beta & \beta & 1 & \delta \\ \gamma & \gamma & \delta & 1 \end{bmatrix}.$$

Note that $\alpha > \beta > \gamma > \delta$, which already hints at a possible ordering of the variables by $R^2$. Computing the $R^2$ yields $R^2(X, 1) \approx 0.59$, $R^2(X, 2) \approx 0.59$, $R^2(X, 3) \approx 0.53$, and $R^2(X, 4) = 0.5$. This means that $X_1$ (the root node) and $X_2$ have the same $R^2$ while, conversely and perhaps surprisingly, the variables $X_2$, $X_3$, and $X_4$, which all have the same fraction of cause-explained variance, have different and descending $R^2$. Even $X_2$ and $X_3$ have different $R^2$ despite having the same fraction of cause-explained variance, and both being neither a root cause nor a leaf node. The $R^2$-sortability of this example is $1/22$. In finite sample settings where the equality of the $R^2$ of $X_1$ and $X_2$ is not exact, sorting by increasing $R^2$ is thus likely to give a causal ordering that is either equal or very close to the inverse of the true causal order. The inverse causal order is markedly different from a random ordering ($\mathbf{v}_{R^2} = 0.5$), and shows that the $R^2$ carry information about the causal ordering. As this example shows, controlling the CEV does not guarantee a $R^2$-sortability close to 0.5, and we recommend evaluating and reporting $R^2$-sortability even when node variances or the cause-explained variance fractions are controlled.