# OpenReview forum: "A Scale-Invariant Sorting Criterion to Find a Causal Order in Additive Noise Models"
_NeurIPS.cc/2023/Conference — NeurIPS 2023 poster_

### Official Review · Reviewer_UFwV · 2023-06-22

**Soundness:** 3 good
**Presentation:** 3 good
**Contribution:** 3 good
**Rating:** 6
**Confidence:** 4

**Summary:**

The paper considers causal inference in the context of additive noise models (ANMs). In particular the paper points out a possible problem in simulation benchmarks for this setup: if the weights of the causal network are not chosen appropriately, the simulation may result in datasets in which identifying the causal ordering can be done with a trivial method which simply sorts the variables by the R^2 when the variable is predicted with other variables. Previously, a similar result has been published by ordering the variables by their variance, but this paper shows that the same outcome can be achieved by sorting the variables by the amount of variance explained, R^2, which is scale-invariant, and hence the problem can't be resolved by simple rescaling.


**Strengths:**

The clarity of the paper is good. The paper demonstrates possible deficiencies in existing benchmarks, which is important. Overall, I found this paper useful, but I also think the significance could be strengthened (see below).

**Weaknesses:**

The paper demonstrates that simulated datasets may have a large R^2 sortability. However, it does not go very far in clarifying how big a problem this is in practice, i.e., it is not really clear how R^2 sortable real-world datasets actually are (there's one example but that's still quite limited). If the real-world datasets actually often are R^2 sortable, then there is no problem. If, on the other hand, they are not R^2 sortable, it would have increased the significance of the paper to provide more concrete suggestions about how simulations should be conducted to in order for them to better correspond to real-world datasets. In real-world datasets the ground-truth may not be easily available, but at least it is easy to calculate the distribution of R^2 values for all variables in a dataset, for multiple datasets, and investigate how the weights in the causal graph simulator should be chosen for the R^2 distribution to be similar to real-world datasets.

The take-home message from the experiments is not always clear. In Fig. 1 the presented method R^2-SortnRegress seems always worse than the previously published Var-Sortnregress. Also, the conclusion in the caption: "R^2Sortnregress performs well it R^2 sortability is high" seems a bit tautological. It is clear that R^2 is scale-invariant unlike the variance based criterion, so why not simulate datasets that accordingly demonstrate its strength? Also, in the real-world data the results don't seem impressive, as the novel R^2 SortnRegress has the worst SID value. (What is the othe value SHD? I didn't find it defined.)

The theory considers a causal chain, where E(log|V|)>0, where V is simulated from the weight distribution, and shows that in this case the variance of the last node in the chain goes to infinity and the amount of variance explained converges to one when the length of the chain increases. It seems that if the expected value of |V| is smaller than 1, then the condition is not satisfied? Indeed, it is easy to imagine that the chain will diverge if we have coefficients that in general are larger than one, but this does not sound very realistic assumption in real-world data sets. It would have been interesting to investigate the distribution of estimated weights in some real-world datasets, to see how commonly this condition holds true in those. The paper says that if the condition did not hold, "detecting statistical dependencies between variables connected by long paths" would be difficult, but I would imagine this to be case with many real-world datasets, so this does not seem a proper reason for not considering such cases.


**Questions:**

Could you show a histogram of R^2 sortability values among simulated datasets on which Fig. 1 was based? Now the R^2 sortability is shown on the x-axis but it is not clear what are the relative amounts of different R^2 values in the simulations.

**Limitations:**

Yes.

---

> ### Author Rebuttal · Authors · 2023-08-09
>
> Thank you for your constructive review and your suggestions on how to better present the strengths of our work.
> We propose the following edits in response and individually answer your questions below.
>
> ---
>
> ### Edit summary
>
> * **Edit 1 (see Figure 2 rebuttal PDF)** We will include the $R^2$-sortability histograms for the settings shown in Figure 1 and 3 in Section A2. For ER graphs, the distribution of $R^2$-sortabilities is close to symmetric, while for SF graphs it has a strong left skew.
>
> * **Edit 2** We will highlight the open nature of real-world $R^2$-sortability (lines 19-21): \
> _Our findings reveal high $R^2$-sortability as an assumption about the data generating process relevant to causal discovery and implicit in the choice of simulation parameters. It should be made explicit, as its prevalence in real-world data is an open question._
>
> * **Edit 3** We will emphasize the impact of $R^2$-sortability on benchmarking practices (line 84): \
> _Characterizing $R^2$-sortability and uncovering its role as a driver of causal discovery performance enables distinguishing between data with different levels of $R^2$-sortability when benchmarking or applying causal discovery algorithms to real-world data._
>
> * **Edit 4 (see Figure 1 in rebuttal PDF)** We have revised Figure 1 (and 3 & 4 likewise) to better show the strengths of our method. We will show results on standardized data to emphasize the scale-invariance of our method. For reference, we visually differentiate the baseline RandomRegress and include Var-SortnRegress on raw data as a dashed line. To simplify the interpretation of trends and uncertainty we will show moving averages using a window of width 0.1 (instead of binning by decile), and show error bars for the 95% confidence interval of the mean.
>
> * **Edit 5** We will clarify the take-home message of the real-world experiment (line 254): \
> _This indicates that, on real-world data, we may not expect to see consistently high $R^2$-sortabilities as much as we do in many simulation settings (see Appendix B.2).
> For benchmarks to be representative of this factor, they should differentiate between settings with different levels of $R^2$-sortability._
>
> * **Edit 6** We will highlight the novelty and usefulness as condition for the divergence of node variances of Equation (5) (line 274): \
> _We introduce the following sufficient condition for weight distributions to result in diverging node variances along causal chains of increasing length:_
>
> ---
>
> ### Answers to your questions
>
> __Histogram of $R^2$-sortabilities (see Figure 1 in rebuttal PDF)__
>
> This is a great idea, thank you. We will include the histograms as described in **Edit 1**.
>
> ---
>
> ### Additional points
>
> **Real-world $R^2$-sortabilities**
>
> We agree that real-world $R^2$-sortabilities are an important open question that cannot be answered by any single dataset (see line 336), and will communicate this more clearly (see **Edit 2**).
> Real-world $R^2$-sortabilities are not known and that is precisely why we think it is important to be aware that current simulations systematically tend to result in high $R^2$-sortabilities.
> Any in causal discovery (e.g. Gaussian vs non-Gaussian, equal vs unequal noise variances, etc.) may or may not hold on different real-world data, which is why they are treated separately in simulations.
> For benchmarking, choosing ANM parameters that result in high $R^2$-sortability amounts to an assumption, which our contribution makes explicit.
> This allows a distinction between data with different $R^2$-sortabilities in the same way we already distinguish between other assumptions to reflect different possible real-world scenarios.
> We agree that this should be spelled out and will do so (see **Edit 3**).
>
> You suggest calculating $R^2$ distributions across multiple datasets and investigating how to match simulations to those values, yet real-world data suiting ANM assumptions are scarce, hence the extensive use of simulations in causal discovery.
> Such a study would have to find datasets, suitable function classes for $R^2$ estimation, and a strategy to adapt simulations (one could change the weights, graph structure, or noise variances) consistent with domain knowledge.
> While this is outside the scope of our work, we share your excitement for such a project and hope our work will inspire studies like these when shared with the community.
>
> **Take-home message of experiments**
>
> We agree that the scale-invariance of $R^2$-sortability should be emphasized more and update accordingly (see **Edit 4**). Thank you for highlighting this. \
> We agree that none of the performances on the real-world dataset are impressive (line 250), which underlines the need for identifying potentially unrealistic benchmark properties such as high $R^2$-sortability. To clarify the take-home message, we will implement **Edit 5** and move the definition of the Structural Hamming Distance (SHD) to the main text.
>
> **Case of $E\log|V|<0$**
>
> Thank you for raising this.
> We can better clarify the role of the condition $E\log|V|>0$, which we prove is sufficient for the divergence of node variances to infinity (see **Edit 6**).
> Indeed, our condition is not satisfied if $E|V|<1$ (Jensen's inequality).
> Conversely,
> $E|V|>1$ does not imply $E\log|V|>0$ (e.g. consider $\text{Unif}(0.1, 2)$),
> and is thus not sufficient for the divergence.
>
> We agree that real-world processes may not have weights that result in diverging variances.
> This is why we think it is important to point out that simulations often rely on parameters that result in high var- and $R^2$-sortability.
> Unfortunately, measuring weights in real-world ANM data (which are scarce) is not straightforward, since they are affected by the arbitrariness of the data scale. \
> We do consider the case of $E\log|V|<0$ in our empirical analyses.
> Figure 2 shows that even $E\log|V|<0$ result in high $R^2$-sortabilities for Scale-free graphs,
> and Section A2.1 shows a setting with small weights and $E\log|V|<0$.

---

> > ### Comment · Reviewer_UFwV · 2023-08-11
> > **Thanks for your replies**
> >
> > Hello,
> > I read your replies and I think they did a good job addressing my concerns. I will take this into account when reconsidering my score (probably during reviewer-AC discussions). I have no further questions. Thanks!

---

> > > ### Author Response · Authors · 2023-08-11
> > >
> > > Thank you for taking the time to read our rebuttal and for your openness to reconsidering your score. We appreciate your constructive feedback, which has improved the quality of our work.
> > >
> > > Please do not hesitate to reach out if any further clarifications are needed.

---

### Official Review · Reviewer_zF5c · 2023-07-03

**Soundness:** 3 good
**Presentation:** 3 good
**Contribution:** 3 good
**Rating:** 7
**Confidence:** 4

**Summary:**

The paper introduces the issue of "$R^2$-sortability" for synthetically generated data used in the evaluation of causal structure learning methods. $R^2$-sortability is a generalization of varsortability, which is invariant to re-scaling (e.g., standardizing) the simulated variables. They show that, using typical simulation parameters, sorting variables based on $R^2$-sortability can give good performance on causal discovery. Finally, they investigate how $R^2$-sortability is influenced by the data generation parameters.

**Strengths:**

**Clarity:** The paper clearly describes the motivation for introducing the concept of $R^2$-sortability, and the experimental results section clearly describes their experimental setup.

**Significance:** There is significant value in papers which describe problems with current methods of evaluation. In particular, since evaluating causal structure learning methods relies heavily on synthetic data, it is important to characterize "artifacts" of synthetic data generation procedures, and make efforts to remove these artifacts. It is also helpful that the authors do some investigation into how $R^2$-sortability is influenced by the synthetic data generation parameters.

**Weaknesses:**

### Experimental Results
It is surprising that, in Section 4.1, the authors do not standardize the data so that *Var-SortnRegress* would perform poorly. It is not clear what message the authors are trying to convey by their experimental setup. I would expect that they would want to show that $R^2$-sortability is still an issue in simulated data, even when varsortability is not an issue.

### Minor issues
*Clarity*: Clarity could be improved in some places.
1. Equation (3) seems to involve the number of paths between pairs of nodes $s$ and $t$, I think it would be more intuitive to write it this way and to also describe why we care about the number of paths rather than just the existence of a path.
2. Equation (3) never decreases when adding an edge. It seems undesirable that the score would always (weakly) favor denser graphs. I see that there is a sparsity penalty in Algorithm 1 - why is it introduced there, instead of earlier?
3. In Section 5, why do you switch from the product of squared edge weights to sum of the logs? It seems motivated by the appeal to the strong law of large numbers, but since you already lower bound the product of squared edge weights by the sum of the logs, then almost sure convergence also holds for the product. Right now, the logs seem ad hoc / unmotivated.
4. When introducing the condition on $\mathbb{E}(\log |V|)$ in Equation (5), you should make it more clear what the conditions means for $P_W$. For example, $P_W = Unif([0.5, 1])$ would not satisfy the condition, would $P_W = Unif([0.5, 2])$ satisfy it?
5. Please use $\mathbb{E}$ for expectations, e.g. in Equation (5). This is typically preferred, but especially important when dealing with graphs where $E$ often denotes edges.

**Questions:**

Please address the issues on clarity raised in the **Weaknesses** section.

### Other suggestions
The authors may find it interesting that previous papers have designed a data generation process to control cause-explained variance, see eg. [1], Section 5 and [2], Section 5. Readers would likely find it helpful if the authors described such a data generation process to ameliorate the issue of $R^2$-sortability.

[1] Agrawal, R., Squires, C., Prasad, N., & Uhler, C. (2021). The DeCAMFounder: Non-linear causal discovery in the presence of hidden variables.
[2] Squires, C., Yun, A., Nichani, E., Agrawal, R., & Uhler, C. (2022). Causal structure discovery between clusters of nodes induced by latent factors.

**Limitations:**

The authors have adequately addressed the limitations of their work.

---

> ### Author Rebuttal · Authors · 2023-08-09
>
> We thank you for your detailed review and for highlighting the significance of our contribution for the causal structure learning community. We propose to make the following edits in response to your review and answer your questions individually below.
>
> ---
>
> ### Edit summary
>
> * **Edit 1 (see Figure 1 in rebuttal PDF)** We have revised Figure 1 (and 3 & 4 likewise) to better show the strengths of our method. We will show results on standardized data to emphasize the scale-invariance of our method. For reference, we visually differentiate the baseline RandomRegress and include Var-SortnRegress on raw data as a dashed line. To simplify the interpretation of trends and uncertainty we will show moving averages using a window of width 0.1 (instead of binning by decile), and show error bars for the 95% confidence interval of the mean.
>
> * **Edit 2** We will add the following explanation to Equation (3) (line 168): \
> _In effect, this is the fraction of directed paths of unique length between any two nodes that satisfy the sortability condition in the numerator._
>
> * **Edit 3 (see Figure 3 in rebuttal PDF)** We will add an appendix section showing an empirical comparison of sortabilities obtained using Equation (3) as is, compared to a version of Equation (3) that only considers the existence of a path. In the experiment we observe strong alignment (Kendall rank coefficients of $0.86$ for $R^2$-sortability and $0.84$ for var-sortability) between the two versions.
>
> * **Edit 4** We will highlight the novelty and usefulness as condition for the divergence of node variances of Equation (5) (line 274) and will motivate the use of the log (line 548): \
> 274: _We introduce the following sufficient condition for weight distributions to result in diverging node variances along causal chains of increasing length:_ \
> 548: _The application of the log in the final step allows us to lower-bound the product by a sum and employ the law of large numbers._
>
> * **Edit 5** We will include the references you suggested in the first paragraph of Section 5 (line 259): \
> _$R^2$-sortability can be mitigated by introducing the assumption of a constant exogenous noise fraction (see for example Agrawal 2021, Squires 2022), which requires non-iid edge weight sampling in simulations._
>
> ---
>
> ### Answers to your questions
>
> **Experimental results**
>
> We agree that the scale-invariance of $R^2$-sortability and -SortnRegress should be emphasized more and will make the changes described in **Edit 1**.
>
> **Minor issues (clarifications)**
>
> *1.* We will add an intuitive description of Equation (3) as outlined in **Edit 2**. Equation (3) builds on the definition by Reisach [2021] in a way that yields their original version of var-sortability as a special case. To address your question, we ran a comparison between the original sortability definition counting paths of different lengths compared to one that only considers the existence of a path and find no qualitative differences. We will include the experiment comparing the sortability variants as described in **Edit 3**.
>
> *2.* Sortability is a measure of a given data generating process and its causal structure. For example, for
>
> $A=N_A$, $\quad B=10 A + N_B$,  $\quad C=0.5 B + N_C \quad$ with graph G1 $(A \to B \to C)$ and $\text{Var}(N_A)=\text{Var}(N_B)=\text{Var}(N_C)=1$ we have $\mathbf{v}_\text{Var}((A,B,C), G1) = 2/3$.
>
> For the data generating process
>
> $D=N_D$, $\quad E=10 D + N_E$, $\quad F=N_F \quad$ with $\text{Var}(N_D)=\text{Var}(N_E)=\text{Var}(N_F)=1$ and graph G2 $(D \to E \quad F)$ we have that $\mathbf{v}_\text{Var}((D,E,F), G2) = 1$.
>
> As this example shows, a data generating process with denser graph can have lower sortability than a DGP with sparser graph (and vice-versa). \
> Algorithm 1 is used to estimate a causal graph from data and the sparsity constraint avoids returning a fully connected graph: after having sorted the variables we regress each node onto its predecessors in this order, and the sparsity constraint prunes superfluous edges.
>
> *3.* The application of the log allows us to make our statement about the divergence of node variances, and Equation 5 introduces a sufficient condition for divergence (the related conditions $\mathbb{E}|V|>1$ or $\mathbb{E}V^2>1$ are not equivalent to our condition as can be seen by the example of $\text{Unif}(0.1, 2)$).
> In particular, if Equation 5 is satisfied then this implies that for increasing chain lengths the sum of log-weights converges to $+\infty$,
> which in turn implies that the product of squared weights converges to $+\infty$.
> If Equation 5 is not satisfied, another condition would be needed to establish convergence of the product to $+\infty$.
> We believe the motivation for applying the log and the role of equation (5) can be made clearer by implementing **Edit 4**. Thank you for pointing this out!
>
> *4.* We will add the values for $P_W = \text{Unif}((-2, -0.5)\cup(0.5, 2))$ ($\approx 0.16$) in section 4.1, and for $P_W = \text{Unif}((-0.5, 0.1)\cup(0.1, 0.5))$ ($\approx -1.29$) in Section A2.1, both with a pointer to Section 5. (Figure 2 shows further weight ranges.)
>
> *5.* Agreed. We will change the notation for the expectation from $E$ to $\mathbb{E}$ throughout the manuscript.
>
> **Suggestions**
>
> Thank you for pointing out the data generating mechanisms in these related works. We will include them as described in **Edit 5** and hope they can serve as a reference point to develop new and more realistic simulation schemes.

---

> > ### Comment · Reviewer_zF5c · 2023-08-13
> >
> > ### Score update
> > I appreciate the author's detailed and well-written response to my review. I was impressed with their quick ability to incorporate feedback, and I believe that their updates to the experiments and their clarifications will further improve the quality of the paper.
> >
> > Taking this into account, I have raised my initial score by one point, from 6 to 7.
> >
> > ---
> > ### Remaining questions
> >
> > **Minor issue 2**: Thank you for clarifying, I somehow missed the change in the denominator. However, the example did not get after my exact concern, which is for a *fixed* data generating process, the score favors denser graphs. In your example, let $G3 = D \to E \to F$, then what is $\mathbf{v}_{\textrm{Var}}((D,E,F), G3)$, i.e. how does the score change when adding $E \to F$? What about when also adding $D \to F$?

---

> > > ### Author Response · Authors · 2023-08-14
> > >
> > > Thank you for acknowledging our rebuttal and how our proposed edits improve the article.
> > > We are grateful for your constructive comments which have helped us refine our submission.
> > >
> > > *Re: sortability and graph density*
> > >
> > > We are not sure if we understand your question on sortabilities when keeping the data-generating process fixed while adding edges.
> > > Below we summarize the relationship between a given DAG and the corresponding sortability, and give a more extensive example.
> > > We hope this answers your question; Please let us know if anything remains unclear and we will be happy to answer.
> > >
> > > ---
> > > ---
> > >
> > > We define sortability for variables $X$ in a graph $G$ (lines 164ff) of an ANM (lines 101ff).
> > > Adding an edge to $G$ to obtain $G'$ necessarily changes the definition of the corresponding variables $X'$, meaning the data-generating process is not fixed across different ground-truth graphs. \
> > > For G' denser than G it may be that
> > > $\\; \mathbf{v}\_\tau(X',G') < \mathbf{v}\_\tau(X,G)$, $\\; \mathbf{v}\_\tau(X',G') > \mathbf{v}\_\tau(X,G)$, or $\\; \mathbf{v}\_\tau(X',G') = \mathbf{v}\_\tau(X,G). \\;$
> > > Adding edges can thus increase or decrease sortability, or leave it unchanged.
> > > To illustrate this point, we give a comprehensive example (using var-sortability for its simpler calculation) below that highlights the relationship between a given DAG and the corresponding variables.
> > >
> > > Let $N_1,N_2,N_3$ be independent random variables with $\\; \text{Var}(N_1)=2$, $\\; \text{Var}(N_2)=2$, $\\; \text{Var}(N_3)=1$.
> > >
> > > __Setting 1__
> > >
> > > For the graph $\\;G : X_1 \to X_2 \quad X_3\\;$
> > > and data-generating process
> > > $\\;X_1:=N_1$, $\\; X_2:=10 X_1 + N_2$, $\\; X_3:=N_3,\\;$,
> > > we have that $\\;\mathbf{v}\_\text{Var}((X_1,X_2,X_3), G) = 1/1 = 1$.
> > >
> > > __Setting 2__
> > >
> > > We now consider the graph $G'$ resulting from adding an edge to $G$ such that
> > > $\\;G' : X'_1 \to X'_2 \to X'_3.\\;$
> > > Adding the edge requires a change in the definition of $X_3$, which changes the data generating process:
> > > $\\;X'_1 = X_1 := N_1$, $\\; X'_2 = X_2 := 10 X_1 + N_2$,  $\\; X'_3 := 0.1 X_2 + N_3, \\;$ which yields $\\;\mathbf{v}\_\text{Var}((X'_1,X'_2,X'_3), G') = 2/3$.
> > >
> > > __Setting 3__
> > >
> > > Adding another edge such that $\\;G'' : X''_1 \to X''_2 \to X''_3, X''_1 \to X''_3\\;$ requires another change of the data generating process.
> > > We show how different edge weights can result in different sortabilities:
> > >
> > > 1. $X''_1 = X_1 := N_1$, $\\; X''_2 = X_2 := 10 X_1 + N_2$,  $\\; X''_3 := 0.1 X_2 - 1.1 X_1 + N_3 \\;$ yields $\\;\mathbf{v}\_\text{Var}((X''_1,X''_2,X''_3), G') = 1/4$.
> > >
> > > 2. $X''_1 = X_1 := N_1$, $\\; X''_2 = X_2 := 10 X_1 + N_2$,  $\\; X''_3 := 0.1 X_2 + 20 X_1 + N_3 \\;$ yields $\\;\mathbf{v}\_\text{Var}((X''_1,X''_2,X''_3), G') = 4/4 = 1$.

---

> > > > ### Comment · Reviewer_zF5c · 2023-08-15
> > > >
> > > > Thanks for clarifying! I see my confusion.
> > > >
> > > > My conception of sortability was closer to scoring functions (such as BIC) used in score-based structure learning algorithms. In that context, for a dataset $\mathcal{D}$, one defines a score $S(\mathcal{D}, G)$ for every graph $G$ and picks the best-scoring graph(s). I was under the impression that the sortability was being used like a scoring function, but I realize that **Algorithm 1** defines a different procedure for finding the graph other than maximizing the $\mathbf{v}_{R^2}$.
> > > >
> > > > It's worth considering clarifying this point, though since it did not confuse any other reviewers, it's probably unnecessary.

---

> > > > > ### Author Response · Authors · 2023-08-18
> > > > >
> > > > > We are happy that our answer could clarify your point and thank you for sharing the source of the confusion.
> > > > > Our approach requires only $d+(d-1)$ regressions, making it computationally more efficient than combinatorial search over all permutations or over all DAGs with $d$ nodes.
> > > > > We agree that we can highlight this aspect more by implementing the edits below.
> > > > >
> > > > > Thank you again for engaging in this conversation!
> > > > >
> > > > > ---
> > > > >
> > > > > - lines 14, 80: ..._computationally efficient_ baseline algorithm...
> > > > > - line 182: _For $d$ variables, $R^2$-SortnRegress fits $d$ regressions to obtain an ordering of the variables by their $R^2$ coefficients, and fits another $d-1$ sparse regressions to determine the presence of edges._

---

### Official Review · Reviewer_iqwe · 2023-07-05

**Soundness:** 3 good
**Presentation:** 4 excellent
**Contribution:** 3 good
**Rating:** 6
**Confidence:** 4

**Summary:**

This paper is concerned with synthetic data generation for causal discovery. The paper explores the scale invariant pattern given by the coefficient of determination $R^2$ that potentially exists in synthetic data benchmarks. The authors present analysis in the case of linear ANMs. They also find out that prior over linear parameters might have significant influence on the $R^2$ sortability.

**Strengths:**

(++) There are many new algorithms being developed for causal discovery, and all of them evaluate on synthetic data as evaluation on real world data is very hard. Hence, any new insights which helps bridge the gap between synthetic data and real data is very timely and welcome.
(++) This work generalises the results of Reisach et al (2021) in giving a general `sortability` criterion. Further, the $R^2$ sortability is more subtle than varsortability as it cannot be just adjusted by renormalisation.
(+) I like the overall presentation and motivation of the problem. The presented analysis for linear ANM and trees is insightful.

**Weaknesses:**

(--) It is not clear how big of an issue is $R^2$ sortability in real world settings. Section 4.2 illustrates it on Sachs dataset. But Sachs dataset is not a linear ANM, and hence has significant model misspecification. So I am unsure how concrete are the conclusions from 4.2. Having said that, I still find the overall contribution useful. For example, if it is the case that there is a $R^2$ sortability issue in a more relevant dataset, one could at least find out to what extent it exists based on the insights of this paper.
(--) A theoretical analysis would have been more helpful. Right now, it is not clear if this is just a problem in linear ANM or can be generalised to nonlinear/nonparametric cases. More so, it might have made more clear whether it is actually the linear parameters which affect it the most in non tree situations (which would be the case in most practical settings).
(--) Unlike the varsortability, which is clear shown that almost all the new causal discovery algorithms being developed exploit it, it is not necessarily the case here. For example, it is not clear whether new optimisation based/ neural network based methods exploit this to perform better. I also don['t think it is necessarily a bad thing to exploit it if it is the case that $R^2$ sortability exists in real world settings (it is yet unclear from this paper, see above point).

**Questions:**

1. Is it possible to generate the synthetic data by first fixing the desired $R^2$, and generate the synthetic data which exactly matches this value?
2. Could you comment on whether this is a potential issue beyond ANMs and linear models?

**Limitations:**

The authors have done a clear job of stating the assumptions under which their conclusions are valid.

---

> ### Author Rebuttal · Authors · 2023-08-09
>
> Thank you for your thoughtful review and for pointing out the impact of our findings on the causal structure learning community. We propose to make the following changes in response to your review and answer your questions individually below.
>
> ---
>
> ### Edit summary
>
> * **Edit 1** We will highlight the open nature of real-world $R^2$-sortability by adapting lines 19-21 (Abstract): \
> _Our findings reveal high $R^2$-sortability as an assumption about the data generating process relevant to causal discovery and implicit in the choice of simulation parameters. It should be made explicit, as its prevalence in real-world data is an open question._
>
> * **Edit 2** We will emphasize the impact of $R^2$-sortability on benchmarking practices 84 (Contribution): \
> _Characterizing $R^2$-sortability and uncovering its role as a driver of causal discovery performance enables distinguishing between data with different levels of $R^2$-sortability when benchmarking or applying causal discovery algorithms to real-world data._
>
> * **Edit 3** We will emphasize the benefit a full theoretical characterization of $R^2$-sortability could provide in line 337 (Discussion): \
> _A complete theoretical characterization of the conditions sufficient and/or necessary for extreme $R^2$-sortability could, if found, help decide where one may hope to exploit it in practice._
>
> ---
>
> ### Answers to your questions
>
> **Generating data with a specific $R^2$-sortability**
>
> Because $R^2$-sortability arises in a complex interplay of simulation parameters there is no systematic way to achieve a target $R^2$-sortability when sampling ANM parameters iid that we know of.
> One may use rejection sampling or update the properties of a given graph to achieve a target $R^2$-sortability with non-iid edge weights (cf. Sections 5 in Agrawal [2021], Squires [2022], as pointed out by reviewer zF5c).
>
> **$R^2$-sortability beyond ANMs and linear models?**
>
> We believe that $R^2$-sortability can be an issue beyond ANMs and linear models, although its impact may vary depending on function class and parameters, and increasing functional complexity along the causal order could reduce or flip $R^2$-sortability.
> In practice, estimating $R^2$ for nonlinear models requires the choice of a suitable regression method.
> Prompted by your question, we ran an experiment using our linear $R^2$-sortability as a first-order approximation on two different nonlinear Gaussian Process simulations used by Zheng [2020] and Reisach [2021].
> Linear $R^2$-sortabilities range from 0.10 to 0.85, indicating that extreme values are possible in non-linear ANMs and making this question an interesting topic for future work.
>
> |graph|SEM|Var-sortability|$R^2$-sortability|
> |---|---|---|---|
> |ER(20, 20)| Additive GP|0.71$\pm$0.13|0.49$\pm$0.18|
> ||GP| 0.66$\pm$0.14| 0.37$\pm$0.13|
> | ER(20, 80)|Additive GP | 0.88$\pm$0.06|0.41$\pm$0.10|
> ||GP| 0.61$\pm$0.13| 0.10$\pm$0.06|
> |SF(20, 20)| Additive GP| 0.83$\pm$0.11|0.85$\pm$0.10|
> ||GP| 0.66$\pm$0.16| 0.52$\pm$0.17|
> | SF(20, 80) | Additive GP| 0.96$\pm$0.02|0.75$\pm$0.14|
> ||GP|0.66$\pm$0.11| 0.23$\pm$0.11|
>
> ----
>
> ### Additional points
>
> **Real-world $R^2$-sortabilities**
>
> We agree that real-world $R^2$-sortabilities are an important open question that cannot be answered by any single dataset (see line 336), and will communicate this more clearly (see **Edit 1**).
> Real-world $R^2$-sortabilities are not known and that is precisely why we think it is important to be aware that current simulations systematically tend to result in high $R^2$-sortabilities.
> Any in causal discovery (e.g. Gaussian vs non-Gaussian, equal vs unequal noise variances, etc.) may or may not hold on different real-world data, which is why they are treated separately in simulations.
> For benchmarking, choosing ANM parameters that result in high $R^2$-sortability amounts to an assumption, which our contribution makes explicit.
> This allows a distinction between data with different $R^2$-sortabilities in the same way we already distinguish between other assumptions to reflect different possible real-world scenarios.
> We agree that this should be spelled out and will do so (see **Edit 2**).
>
> **Theoretical analysis**
>
> $R^2$ depends on a complex interplay of weights, noise parameters, and graph structure, which greatly complicates the development of a general theoretical characterization of $R^2$-sortability for generic and flexible parameter sampling schemes.
> While it is outside the scope (and page limit) of our contribution, we strongly share your view that more research on the conditions for the emergence of $R^2$-sortability would be beneficial, and hope that sharing our work with the NeurIPS community will inspire research leading to new theoretical insights.
> We will state the potential benefit of such a result through **Edit 3**.
>
> **Meaning of $R^2$-sortability for existing algorithms**
>
> As a property of the data, $R^2$-sortability affects the baseline performance that can be achieved and is thus relevant to the interpretation of the results obtained by causal discovery algorithms (we do not claim that other causal discovery algorithms exploit $R^2$-sortability).
> For example, the same causal discovery performance may be less impressive if the $R^2$-sortability on a dataset is 0.95 rather than 0.5.

---

> > ### Comment · Area_Chair_7yhE · 2023-08-20
> >
> > Dear Reviewer iqwe,
> >
> > The authors have provided a response to your review comments. Could you see whether your concerns were properly addressed by the authors' response, or at least acknowledge you read it?
> >
> > Many thanks,
> >
> > The AC

---

> > > ### Comment · Reviewer_iqwe · 2023-08-20
> > > **Acknowledgement of the Rebuttal**
> > >
> > > Thanks a lot for answering my questions and highlighting the possible changes to the paper. I have no further questions. I will maintain my positive rating of the paper.

---

### Official Review · Reviewer_j7gp · 2023-07-06

**Soundness:** 3 good
**Presentation:** 3 good
**Contribution:** 3 good
**Rating:** 7
**Confidence:** 3

**Summary:**

This paper proposes an interesting extension of the var-sortability approach proposed by Reisach et al. (2021), denoted R2-sortability. While var-sortability measures the agreement between the causal order and the order of increasing marginal variance, R2-sortability measures the agreement between causal ordering and the order of the explainable fraction of a variable’s variance (as captured by the coefficient of determination R2). Contrary to var-sortability, the R2-sortability approach is a scale-invariant approach which can be applied even when the data scale is arbitrary.

The paper proposes a baseline algorithm, “R2-SortnRegress” which can match, and even exceed, the performance of established causal discovery methods such as the PC and FGES algorithms. Comparisons against these baselines (and the Var-SortnRegress and RandomRegress baselines) were performed using synthetic data generated from random and scale-free graphs using Gaussian noise data (with noise standard deviations and edge weights i.i.d. sampled from uniform distributions). The paper also presents a brief real data comparison using a single dataset.

The paper provides a mathematical analysis of the influence of the weight distribution on R2 in the simple setting of causal chains, as well as, an empirical evaluation on more complex random graphs, underscoring the role of the weight distribution as a driver of R2.
The analyses presented in the paper illustrate the need to make decisions on all ANM parameters in simulations.


**Strengths:**

This is a well written and interesting paper. The approach is novel and its need is well motivated. The proposed approach is sound and well described in the text.

**Weaknesses:**

I only have a few very minor comments. The notation is sometimes a little inconsistent. For instance, $\sigma$ and $\phi$ are used interchangeably to represent the standard deviation. Also, in Algorithm 1, $\sigma$ is also used to represent the permutation that sorts $\pi$ in ascending order.


**Questions:**

Why did the paper only select the PC and FGES as traditional causal discovery algorithms in their comparisons? (The Reisach et al. (2021) paper includes other popular methods.)

Also, why the synthetic data experiments only report SID, while the real data illustration reports SID and SHD?


**Limitations:**

Yes.

---

> ### Author Rebuttal · Authors · 2023-08-09
>
> We thank you for your thorough and positive review.
> We propose to make the following edits in response to your review and answer your questions individually below.
>
> ---
>
> ### Edit summary
>
> * **Edit 1** We will include DirectLiNGAM (Shimizu 2021) as an additional algorithm in our comparisons in Appendix A3, Figure 5 where we consider non-Gaussian noise, among other settings.
> * **Edit 2** We will add SHD results for the settings shown in Figure 1 and Figure 3 as a new subsection in Appendix A2 and refer to them in Section 4.1.
> * **Edit 3** We will replace $\sigma$ in the algorithm and align the use of $\phi$ and $\sigma$.
>
> ---
>
> ### Answers to your questions
>
> **Comparison to other algorithms**
>
> Following the findings by Reisach [2021] we decided to include Var-SortnRegress as a baseline representative of the performance of scale-sensitive algorithms given the large and growing number of such algorithms (Vowels 2021).
> In Section A3 we also include the _Top-Down_ algorithm as an additional method that exploits another type of assumption (equal noise variances), which performs largely similar to _Var-SortnRegress_. \
> We will further add the DirectLiNGAM algorithm as described in **Edit 1**.
> DirectLiNGAM highlights the specificity of the non-Gaussian setting by outperforming the other methods for non-Gaussian noise but performing badly otherwise, in line with theory.
> We may be able to include another algorithm for the camera-ready version of the paper if there is one that would provide a substantially different aspect or perspective and thus help highlight the main takeaway of the comparison?
>
> **Addition of SHD results (qualitatively the same as SID results)**
>
> Thank you for the suggestion. We agree that the SHD results may be interesting to readers and will add them as described in **Edit 2**.
> The trends in terms of SHD are the same as in the SID results. $R^2$-SortnRegress improves with increasing $R^2$-sortability, and absolute performances are better on SF graphs than on ER graphs.
>
> ---
>
> ### Additional points
>
> **Notation** — Thank you for pointing this out, we will address it (cf. **Edit 3**).

---

> > ### Comment · Reviewer_j7gp · 2023-08-19
> >
> > Thank you for your responses. The inclusion of DirectLiNGAM in the non-gaussian setting makes sense. I have no further questions and will maintain my overall score of 7.

---

### Author Rebuttal · Authors · 2023-08-09

Dear All,

We thank you for the review of our submission, valuable suggestions, and recognition of our work's significance and timeliness for the causal discovery community.
All reviews seem to agree on the good technical soundness (4x good)
and presentation (1x excellent, 3x good) of the overall contribution (3x good, 1x fair) our paper makes by characterizing a novel pattern implicit in the parameters of additive noise models that can strongly affect causal discovery.

We have revised our manuscript carefully and made improvements based on all reviewers' comments:

* _Updated Figures._ We have improved existing figures and created new figures to complement the presentation of our experimental results and better highlight some of the strengths of our work (scale-invariance of $R^2$-sortability and -SortnRegress, distribution of $R^2$-sortabilities, robustness to different ways of counting sortable node pairs).
* _Revisions._ We have rephrased or added individual sentences to clarify explanations and include two new references.

We summarize these edits below and provide detailed responses to all reviewers in our individual replies.

We are looking forward to discussing our work with you and hope our replies answer your questions and help in your assessment of our submission.
We will continue to revise our manuscript if further suggestions arise during the author-reviewer discussion.

---

__Updated Figures__

- __see Figure 1 in attached PDF__ [zF5c, UFwV] (Figure 1, Section 4.1). \
We have revised Figure 1 (and 3 & 4 likewise) to better show the strengths of our method. We will show results on standardized data to emphasize the scale-invariance of our method. For reference, we visually differentiate the baseline RandomRegress and include Var-SortnRegress on raw data as a dashed line. To simplify the interpretation of trends and uncertainty we will show moving averages using a window of width 0.1 (instead of binning by decile), and show error bars for the 95% confidence interval of the mean.

- __see Figure 2 in the attached PDF__ [UFwV] (Appendix A2). \
We will include the $R^2$-sortability histograms for the settings shown in Figure 1 and 3 in Section A2. For ER graphs, the distribution of $R^2$-sortabilities is close to symmetric, while for SF graphs it has a strong left skew.

- __see Figure 3 in the attached PDF__ [zF5c] (new Appendix C). \
We will add an appendix section showing an empirical comparison of sortabilities obtained using Equation (3) as is, compared to a version of Equation (3) that only considers the existence of a path. In the experiment we observe strong alignment (Kendall rank coefficients of $0.86$ for $R^2$-sortability and $0.84$ for var-sortability) between the two versions.

- [j7gp] (Figure 5, Appendix A3). \
We will include DirectLiNGAM (Shimizu 2021) as an additional algorithm in our comparisons in Appendix A3, Figure 5 where we consider non-Gaussian noise, among other settings.
We will also add the SHD results corresponding to Figures 1 and 3 in A2 (the trends are the same as for SID).


__Revisions__

* We will highlight the open nature of real-world $R^2$-sortability [iqwe, UFwV] (lines 19-21, Abstract). \
_Our findings reveal high $R^2$-sortability as an assumption about the data generating process relevant to causal discovery and implicit in the choice of simulation parameters. It should be made explicit, as its prevalence in real-world data is an open question._

* We will emphasize the impact of $R^2$-sortability on benchmarking practices [iqwe, UFwV] (line 84, Contribution). \
_Characterizing $R^2$-sortability and uncovering its role as a driver of causal discovery performance enables distinguishing between data with different levels of $R^2$-sortability when benchmarking or applying causal discovery algorithms to real-world data._

* We will add the following explanation to Equation (3) [zF5c] (line 168, Section 3.1). \
_In effect, this is the fraction of directed paths of unique length between any two nodes that satisfy the sortability condition in the numerator._

* We will clarify the implications of our findings on real-world data for simulations [UFwV] (line 254, Section 4.2). \
_This indicates that, on real-world data, we may not expect to see consistently high $R^2$-sortabilities as much as we do in many simulation settings (see Appendix B.2).
For benchmarks to be representative of this factor, they should differentiate between settings with different levels of $R^2$-sortability._

* We will include the suggested references [zF5c] (line 259, Section 5). \
_$R^2$-sortability can be mitigated by introducing the assumption of a constant exogenous noise fraction (see for example Agrawal 2021, Squires 2022), which requires non-iid edge weight sampling in simulations._

* We will highlight the novelty and usefulness as condition for the divergence of node variances of Equation (5) (line 274) [zF5c, UFwV] (line 274, Section 5.1). \
_We introduce the following sufficient condition for weight distributions to result in diverging node variances along causal chains of increasing length:_

* We will emphasize the benefit a full theoretical characterization of $R^2$-sortability could provide [iqwe] (line 337, Section 6). \
_A complete theoretical characterization of the conditions sufficient and/or necessary for extreme $R^2$-sortability could, if found, help decide where one may hope to exploit it in practice._

---

### Decision · Program_Chairs · 2023-09-21

**Decision:**

Accept (poster)

**Comment:**

This paper is concerned with finding causal order among the given variables according to the additive noise model. It demonstrates a possible R2 sortability problem in simulation benchmarks. The finding is interesting and has immediate implications for designing simulations to evaluate causal discovery methods.